# mTORC1 signaling facilitates differential stem cell differentiation to shape the developing murine lung and is associated with mitochondrial capacity

Kuan Zhang ®[1], Erica Yao[1], Ethan Chuang[1], Biao Chen[1], Evelyn Y. Chuang[1] & Pao-Tien Chuang ®[1] ✉

Formation of branched organs requires sequential differentiation of stem cells. In this work, we find that the conducting airways derived from SOX2+ progenitors in the murine lungs fail to form without mTOR complex 1 (mTORC1) signaling and are replaced by lung cysts. Proximal-distal patterning through transitioning of distal SOX9+ progenitors to proximal SOX2+ cells is disrupted. Mitochondria number and ATP production are reduced. Compromised mitochondrial capacity results in a similar defect as that in mTORC1-deficient lungs. This suggests that mTORC1 promotes differentiation of SOX9+ progenitors to form the conducting airways by modulating mitochondrial capacity. Surprisingly, in all mutants, saccules are produced from lung cysts at the proper developmental time despite defective branching. SOX9+ progenitors also differentiate into alveolar epithelial type I and type II cells within saccules. These findings highlight selective utilization of energy and regulatory programs during stem cell differentiation to produce distinct structures of the mammalian lungs.

Developmental processes are executed through successful implementation of cellular functions in distinct cell types and in a spatially and temporally specific manner. These cellular functions underlie key processes such as cell proliferation, migration, differentiation and morphogenesis. Essential cellular function relies on energy supply. Thus, it is assumed that all developmental processes are energy dependent. However, it is unclear if different cell types and cellular processes display a similar level of energy dependence in vivo. Answers to this question will provide insight into how energy is utilized at tissue and organismal levels during development.

Generation of the elaborate respiratory tree in mammals requires the production of an appropriate number of cells and acquisition of cellular properties that enable cell migration, differentiation and morphological changes[1-4]. Extensive studies have uncovered genes and pathways and the cellular events that control many aspects of these processes[5-8]. The lungs undergo multiple rounds of stereotypic branching[9] to increase the total surface of gas exchange. Once lung branches are produced, SOX2+ cells demarcate the proximal airway while SOX9+ cells are distributed in the airways distal to the SOX2+ domain[10]. Expansion of the distal SOX9+ cell population is proposed to give rise to proximal SOX2+ cells through changes in the transcription program. In particular, cessation of Sox9 expression is largely concomitant with Sox2 activation and subsequent formation of the conducting airways. As a result, the conducting airways extend distally due to the sequential addition of newly generated SOX2+ cells, which differentiate into various cell types including club cells, ciliated cells and goblet cells. The distal SOX9+ progenitors subsequently generate saccules and differentiate into alveolar epithelial type I (AT1) and

[1]Cardiovascular Research Institute, University of California, San Francisco, CA, USA. ✉e-mail: pao-tien.chuang@ucsf.edu

type II (AT2) cells. The alveolus[11–13], the unit of gas exchange, forms within saccules. Little is known about how energy utilization impacts distinct processes during lung development.

mTOR complex 1 (mTORC1) controls many essential cellular functions[14,15]. It is indispensable for cell growth, cell proliferation and cell cycle progression[16]. mTORC1 is also a sensor of energy and nutrients and controls biomass accumulation and multiple aspects of cell metabolism including anabolic and catabolic processes[17]. mTORC1 modulates ATP production via regulating the capacity of the mitochondria[18,19]. *Rptor* (*Raptor, rapamycin-sensitive regulatory associated protein of mTOR*)[20] encodes an essential component of mTORC1, which includes the RPTOR scaffold protein, the mTOR kinase and several other proteins[21]. How mTORC1 signaling regulates lung organogenesis has not been reported. Manipulating mTORC1 or mitochondrial function thus provides a means to modulate energy utilization during organ formation in animals.

In this work, we have employed lung development in mice as a system to decipher energy dependence in different cell types and cellular processes in vivo. Through control of mTORC1 signaling and mitochondrial capacity during lung development, we have discovered energy dependence for distinct cellular processes to form lung branches and saccules. The transition of the distal SOX9+ progenitors to proximal SOX2+ progenitors to produce the conducting airways is vulnerable to energy reduction. By contrast, morphogenesis of the distal SOX9+ progenitors to generate saccules and differentiation into alveolar epithelial cells within saccules are more resistant to energy deficiency. Through RNA-Seq we have identified the downstream effectors (including Hedgehog signaling) of the mTORC1–mitochondrial axis in forming the conducting airways. These findings suggest that a given cellular process in a distinct environment in vivo displays different energy requirements. This approach provides a way to reveal how energy utilization impacts cellular processes and classify cellular processes in terms of energy utilization.

## Results

### Inactivation of mTORC1 signaling in the lung epithelium disrupts branching morphogenesis

To study the effect of energy utilization on lung branching, we eliminated *Raptor* (*Rptor*), a key component of mTORC1, in the murine lung epithelium. We produced *Rptor^f/f^; Shh^Cre/+^* mice, in which broad expression of *Shh^Cre^*[22] in the lung epithelium converted a floxed (f) allele of *Rptor* (*Rptor^f^*)[20] to a null allele (Supplementary Fig. 1c). As a result, mTORC1 function was disrupted. Consequently, the cellular processes controlled by mTORC1 were perturbed. For instance, mTORC1-dependent phosphorylation of ribosomal protein S6 (RPS6)[23] was reduced in the lung epithelium of *Rptor^f/f^; Shh^Cre/+^* mice at 12.5 *days post coitus* (*dpc*) (Fig. 1a, b).

While development of the trachea and main stem bronchi was normal in *Rptor^f/f^; Shh^Cre/+^* lungs, branching morphogenesis was compromised (Supplementary Fig. 1a, b, d). Defective branching was first observed at 12.5 *dpc* and continued throughout lung development (Fig. 1c, d). In control lungs at 11.5 *dpc*, the five main branches (right cranial [RCr], right middle [RMd], right caudal [RCd], right accessory [RAc] and left [L]) were established to generate the five lobes in adult lungs[9]. Daughter branches have emerged from the five main branches in control lungs at 12.5 *dpc* (Fig. 1c, d). By contrast, *Rptor^f/f^; Shh^Cre/+^* lungs maintained the rudimentary pattern of five main branches without new daughter branches at 12.5 *dpc* (Fig. 1c, d). Multiple solid sphere-shaped structures formed at the distal part of *Rptor^f/f^; Shh^Cre/+^* lungs (Fig. 1c, d) and these animals died soon after birth due to respiratory failure. These results suggest that mTORC1 plays an

essential role in controlling lung branching. Analysis of the branching defects in *Rptor^f/f^; Shh^Cre/+^* mice provides a unique opportunity to explore how energy utilization impacts specific developmental processes.

To investigate whether loss of mTORC1 altered cellular behaviors and thus defective branching, we probed the properties of epithelial cells. Cell proliferation assessed by EdU incorporation[24] was unaffected in *Rptor^f/f^; Shh^Cre/+^* lungs at 12.5–13.5 *dpc* (Supplementary Fig. 2). mTORC1-deficient epithelial cells in either the proximal SOX2+ or distal SOX9+ domain had a similar rate of EdU incorporation in comparison with control lungs. This observation suggests that the branching defect in *Rptor*-deficient lungs was not due to reduced cell proliferation. We speculate that production of new lung epithelial cells in the absence of stereotypic branching leads to the formation of sphere-shaped structures and subsequently cysts in *Rptor^f/f^; Shh^Cre/+^* lungs.

We noticed reduced expression of several markers at the apical surface of epithelial cells in *Rptor^f/f^; Shh^Cre/+^* lungs at 13.5 *dpc*. For instance, the actin cytoskeleton (labeled by phalloidin)[25], pMLC2 (indicative of mechanical force production)[26] and PKCζ (apical epithelial cell marker)[27] were preferentially concentrated at the apical surface of epithelial cells in control lungs. Instead, phalloidin, pMLC2 and PKCζ levels were reduced at the apical surface of *Rptor^f/f^; Shh^Cre/+^* lungs (Fig. 1e, g and Supplementary Fig. 3). These results suggest that removal of mTORC1 was associated with loss of apical-basal polarity and mechanical force production in epithelial cells. These defects could contribute to abnormal branching morphogenesis in *Rptor^f/f^; Shh^Cre/+^* lungs.

### The SOX9+ population in the distal airway fails to generate SOX2+ cells in the absence of mTORC1

Since lung branching is disrupted in *Rptor^f/f^; Shh^Cre/+^* lungs, we investigated if specification of the proximal-distal airway was perturbed in the absence of mTORC1 signaling. This could be the molecular basis of defective branching. During lung branching, patterning of the airway smooth muscle is closely coordinated with patterning of the airways. Alpha smooth muscle actin (SMA)[28] is highly expressed in mesenchymal cells along the epithelial tube except at the tip of lung branches in control lungs. SMA expression in *Rptor^f/f^; Shh^Cre/+^* lungs was largely absent along the stratified proximal epithelium and was only detected in the most proximal region (Fig. 1f, h). This suggests that proximal-distal patterning of airway smooth muscles is disrupted in *Rptor*-deficient lungs. Moreover, proximal-distal patterning of blood vessels[29], which follow airway branching, was also perturbed in *Rptor*-deficient lungs (Fig. 2a). These findings suggest that removal of epithelial *Rptor* disturbs proximal-distal patterning of all structures, including the airways, smooth muscles and blood vessels.

To affirm this notion, we examined the distribution of SOX2+ and SOX9+ epithelial cells during lung development. The segregation of SOX2+ and SOX9+ cells was already apparent at 11.5 *dpc* in *Rptor^f/f^; Shh^Cre/+^* lungs when no apparent structural abnormality could be discerned between control and *Rptor^f/f^; Shh^Cre/+^* lungs (Fig. 2b, c, d and Supplementary Fig. 4a). At this stage, SOX2+ cells were detected in the five main branches derived from the two primary buds in control lungs. By contrast, SOX2+ cells were absent in the corresponding lung branches in *Rptor^f/f^; Shh^Cre/+^* lungs. As development proceeded, the SOX2+ domain remained confined to the trachea and main stem bronchi of the mutant lungs (Fig. 2b, e, f and Supplementary Fig. 4b). This result suggests that the program that produces SOX2+ cells from SOX9+ progenitor cells is not turned on in the absence of mTORC1. As a result, the conducting airways that are derived from SOX2+ cells fail to form in *Rptor*-deficient lungs. It also implies that loss of mTORC1 has a

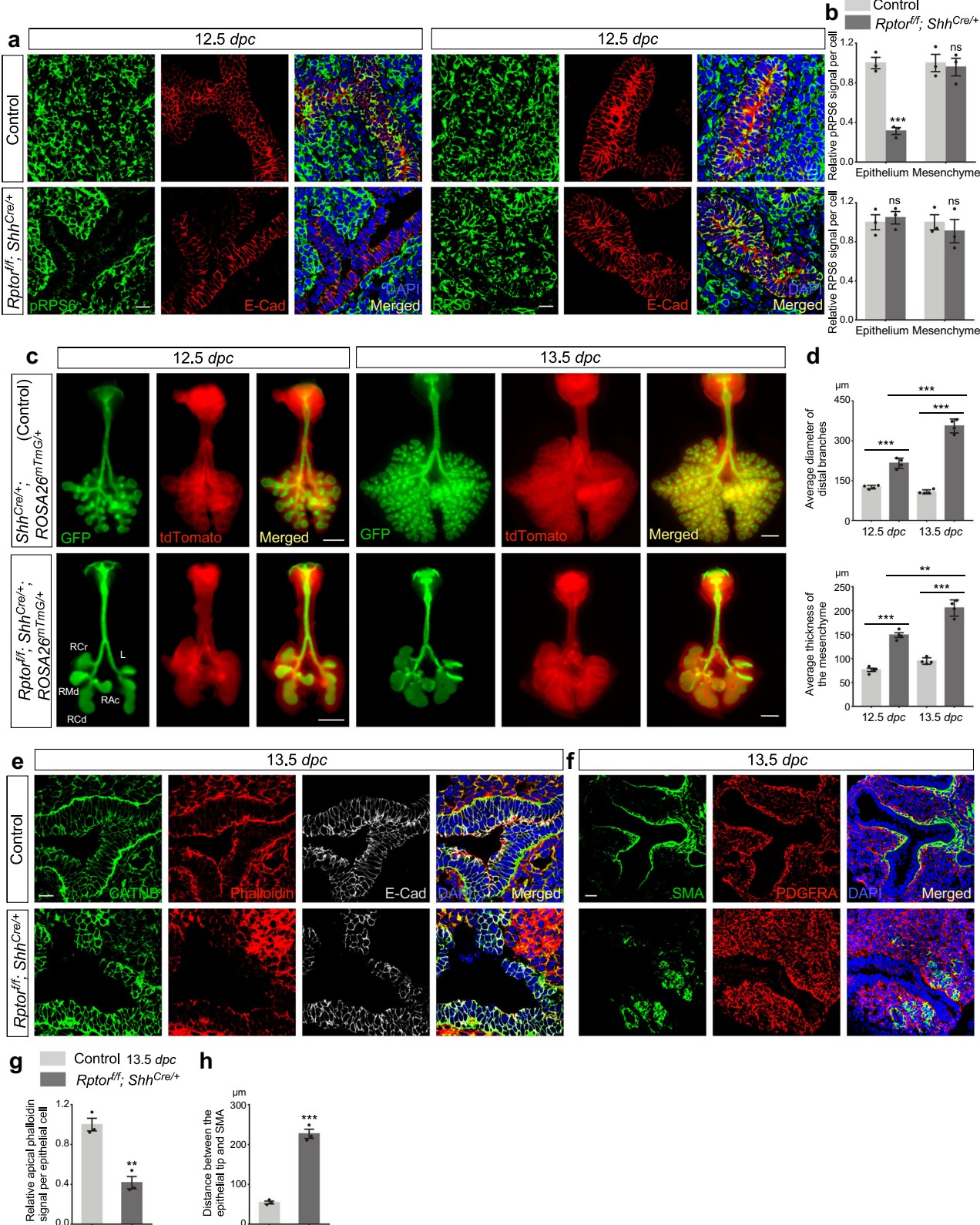

fundamental impact on the acquisition of cellular properties in SOX9+ progenitor cells.

## SOX9+ cells lacking mTORC1 still form saccules and differentiate into AT1 and AT2 cells

Branching morphogenesis is followed by the formation of saccules at the end of lung branches and specification of cell types in both the proximal and distal airways. In control lungs at 16.5 dpc, the distal SOX9+ progenitors produced saccules, which were composed of AT1 (AQP5+/T1α+/HOPX+) and AT2 (SPC+) cells. By contrast, cavitation of solid sphere-shaped structures in the distal regions of Rptor^f/f; Shh^Cre/+ lungs produced cysts, which became apparent at 16.5 dpc (Fig. 3a and Supplementary Fig. 5). Rptor^f/f; Shh^Cre/+ lungs at 18.5 dpc consisted of a few large cysts (Fig. 3b) without forming the conducting airways.

**Fig. 1 | *Rptor* controls mammalian lung branching. a** Immunostaining of lung sections collected from *Rptor^{f/f}; Shh^{Cre/+}* mice and control littermates at 12.5 *days post coitus* (*dpc*). E-Cadherin (E-Cad; CDH1) labels all epithelial cells, while phosphorylated RPS6 (pRPS6) is a marker of mTORC1 activity. A sharp reduction in pRPS6 signal in the lung epithelium indicates efficient recombination of the *Rptor* flox allele (*Rptor^f*) by *Shh^{Cre}*. Scale bar = 10 μm. **b** Quantification of the relative signal of pRPS6 and RPS6 in the lung epithelium and mesenchyme (*n* = 3 for each group) at 12.5 *dpc*. Five views per lung (a total of 15 views) were used for quantification. The signal per cell was calculated as follows. The signal density in control and mutant lungs was measured using ImageJ, and then divided by the corresponding number of cells. **c** Whole-lung imaging of dissected lungs from *Shh^{Cre/+}; ROSA26^{mTmG/+}* (control) and *Rptor^{f/f}; Shh^{Cre/+}; ROSA26^{mTmG/+}* mice at 12.5 and 13.5 *dpc*. GFP was activated from the *ROSA26^{mTmG}* allele in all lung epithelial cells by *Shh^{Cre}*; tdTomato marked all non-epithelial cells. Epithelial branching was perturbed at 12.5 *dpc*. A thickened layer of non-epithelial cells (tdTomato^+) was observed in *Rptor^{f/f}; Shh^{Cre/+}; ROSA26^{mTmG}* lungs. Scale bar = 0.5 mm. **d** Quantification of the average diameter of distal branches (GFP^+) and the average thickness of the mesenchyme (tdTomato^+) surrounding the distal buds of *Shh^{Cre/+}; ROSA26^{mTmG/+}* (control) and *Rptor^{f/f}; Shh^{Cre/+}; ROSA26^{mTmG/+}* lungs at 12.5 and 13.5 *dpc* (*n* = 4 for each group). At least 5 buds per lung were used for measurement by ImageJ. The increased diameter and thickness in the mutant lungs from 12.5 to 13.5 *dpc* are consistent with halted branching morphogenesis. **e** Immunostaining of lung sections collected from *Rptor^{f/f}; Shh^{Cre/+}* mice and control littermates at 13.5 *dpc*. CATNB (CTNNB1) is expressed in all cells while E-Cad only labels epithelial cells. Phalloidin binds to F-actin and marks the actin cytoskeleton. A decreased apical actin cytoskeleton was observed in *Rptor^{f/f}; Shh^{Cre/+}* lungs. Scale bar = 10 μm. **f** Immunostaining of lung sections collected from *Rptor^{f/f}; Shh^{Cre/+}* mice and control littermates at 13.5 *dpc*. SMA (ACTA2) and PDGFRA labeled airway and vascular smooth muscle cells, where PDGFRA labels all smooth muscle cells while SMA largely marks smooth muscle cells in the proximal airway. Many PDGFRA^+ cells with reduced SMA expression were found in *Rptor^{f/f}; Shh^{Cre/+}* lungs. Scale bar = 25 μm. **g** Quantification of the relative apical phalloidin signal per epithelial cell (n = 3 for each group). Five views per lung (a total of 15 views) were quantified as in (**b**). **h** Quantification of the distance between the epithelial tip and the position of SMA expression (*n* = 3 for each group). Five views per lungs (a total of 15 views) were used. The epithelial tip was identified by DAPI. All values are mean ± SEM. (**) *p* < 0.01; (***) *p* < 0.001; ns not significant (two-tailed, unpaired Student's *t* test). Source data are provided as a Source Data file.

Unexpectedly, despite lack of branching morphogenesis, the cysts in *Rptor^{f/f}; Shh^{Cre/+}* lungs contained saccule-like structures on the surface (Fig. 3c, d). At 15.5 *dpc*, structures with focal SOX9 expression could be observed on the surface of *Rptor*-deficient cysts (Supplementary Fig. 6a, b). These SOX9^+ foci coalesced into irregular patches and developed into saccule-like structures at 16.5 *dpc* (Fig. 3c and Supplementary 6c, d, e). These saccule-like structures on the cystic surface of *Rptor^{f/f}; Shh^{Cre/+}* lungs further branched and became interconnected, reminiscent of saccule formation in control lungs (Supplementary Fig. 7). To confirm that the saccule-like structure in *Rptor^{f/f}; Shh^{Cre/+}* lungs was derived from the distal lung epithelium, we examined marker expression in these regions. Indeed, SOX2 was absent in saccule-like structures (Fig. 4a, b, e, h, i and Supplementary Fig. 6).

We then asked whether production of AT1 and AT2 cells from the SOX9^+ domain was affected in *Rptor^{f/f}; Shh^{Cre/+}* lungs. We found that SOX9^+ saccule-like structures were populated by AT2 and AT1 cells (Figs. 3c, d and 4c, g and Supplementary Fig. 8c, d). SPC^+ AT2 cells were interspersed within a layer of AT1 cells and preferentially distributed at the tip of saccules in a pattern similar to that in control saccules. By contrast, cell types in the conducting airways such as club cells (CC10^+) and ciliated cells (Ac-tub^+) were absent in *Rptor^{f/f}; Shh^{Cre/+}* lungs (Fig. 4d, f and Supplementary Fig. 8a, b). These findings suggest that while production of SOX2^+ cells from the SOX9^+ domain was perturbed, SOX9^+ cells were capable of forming saccules and differentiating into AT1 and AT2 cells.

### RNA-Seq analysis of mTORC1-deficient lungs reveals perturbed pathways involved in branching morphogenesis

To gain additional insight into how mTORC1 controls lung branching, we collected lungs from control and *Rptor^{f/f}; Shh^{Cre/+}* mice at 11.5 *dpc* and performed RNA-Seq analysis[30] to probe the changes in transcriptomes in the absence of mTORC1 (Fig. 5a, b, c). Pathway analysis revealed perturbation of the tubular branching process and cell junction formation (Fig. 5b). Other pathways that control extracellular matrix organization, mesenchymal cell differentiation, cytosolic ribosome organization, apicolateral plasma membrane and basement membrane organization were also affected (Fig. 5c). Moreover, genes involved in the glycolytic pathway were also altered (Fig. 5a). This suggests that disturbed energy utilization in *Rptor*-deficient lungs is linked to cellular processes that control lung branching. RNA-Seq analysis at 11.5 *dpc* also revealed a sharp reduction in *Sox2* levels while the expression levels of *Sox9* were unaltered. This is consistent with a blockade in producing SOX2^+ cells from SOX9^+ progenitors in the absence of mTORC1.

### Mitochondrial capacity is reduced in mTORC1-deficient lungs

Our previous work uncovered a role of mTORC1 in controlling mitochondrial capacity during alveolar development[31]. This prompted us to investigate if ablation of mTORC1 signaling during lung branching also impacted mitochondrial capacity and thus branching morphogenesis. We found that the relative ratio of mitochondrial DNA (mtDNA)[32] to nuclear DNA (nDNA) was reduced in *Rptor^{f/f}; Shh^{Cre/+}* lungs compared to controls (Fig. 5d, e), suggesting perturbed mitochondrial function. Indeed, relative ATP production was reduced in *Rptor^{f/f}; Shh^{Cre/+}* lungs compared to controls (Fig. 5f). We also determined mitochondrial mass in *Rptor*-deficient lung epithelium using MPC1 (mitochondrial pyruvate carrier 1)[33] or MTCO1 (mitochondrially encoded cytochrome c oxidase I)[34] as markers that label mitochondria in vivo. Mitochondrial mass was greatly decreased in *Rptor^{f/f}; Shh^{Cre/+}* lungs (Fig. 5g, h, i, j and Supplementary Fig. 9). These findings support a model in which mTORC1 signaling controls lung branching and saccule formation in part through its influence on mitochondrial capacity.

### Reduced mitochondrial function leads to branching defects but preserves saccule formation

Our observations suggest that a main target of mTORC1 signaling during lung branching and saccule formation is the mitochondria. We tested if perturbation of mitochondrial function also generated branching defects. To this end, we created *Tfam^{f/f}; Shh^{Cre/+}* and *Cox10^{f/f}; Shh^{Cre/+}* mice. *Tfam* encodes a major transcription factor for mitochondria while *Cox10* encodes an essential component of the respiratory chain in mitochondria. Likewise, expression of *Shh^{Cre}* in the lung epithelium converted a floxed allele of *Tfam* (*Tfam^f*)[35] or *Cox10* (*Cox10^f*)[36] into a null allele (Supplementary Fig. 10a). Loss of either *Tfam* or *Cox10* is expected to disrupt mitochondrial function. A global reduction of MTCO1 in lung epithelium of *Tfam^{f/f}; Shh^{Cre/+}* and *Cox10^{f/f}; Shh^{Cre/+}* mice indicated efficient removal of *Tfam* and *Cox10* by *Shh^{Cre}* (Fig. 6a, b, c, d). A reduction in MTCO1 signal in *Tfam*-deficient cells was due to reduced transcription of MTCO1. By contrast, MTCO1 protein becomes unstable in the absence of COX10, resulting in decreased MTCO1 immunoreactivity. Moreover, the relative mtDNA/nDNA ratio was decreased in *Tfam^{f/f}; Shh^{Cre/+}* lungs (Fig. 6e) but not in *Cox10^{f/f}; Shh^{Cre/+}* lungs (Fig. 6f). Relative ATP production was compromised in *Tfam^{f/f}; Shh^{Cre/+}* and *Cox10^{f/f}; Shh^{Cre/+}* lungs (Fig. 6g). These findings indicate that loss of epithelial *Tfam* or *Cox10* reduces mitochondrial capacity.

We found that lungs from *Tfam^{f/f}; Shh^{Cre/+}* and *Cox10^{f/f}; Shh^{Cre/+}* mice displayed defective branching morphogenesis (Fig. 6h and Supplementary Fig. 10b). The defects appeared slightly later than those in *Rptor^{f/f}; Shh^{Cre/+}* lungs. Abnormal branching could be

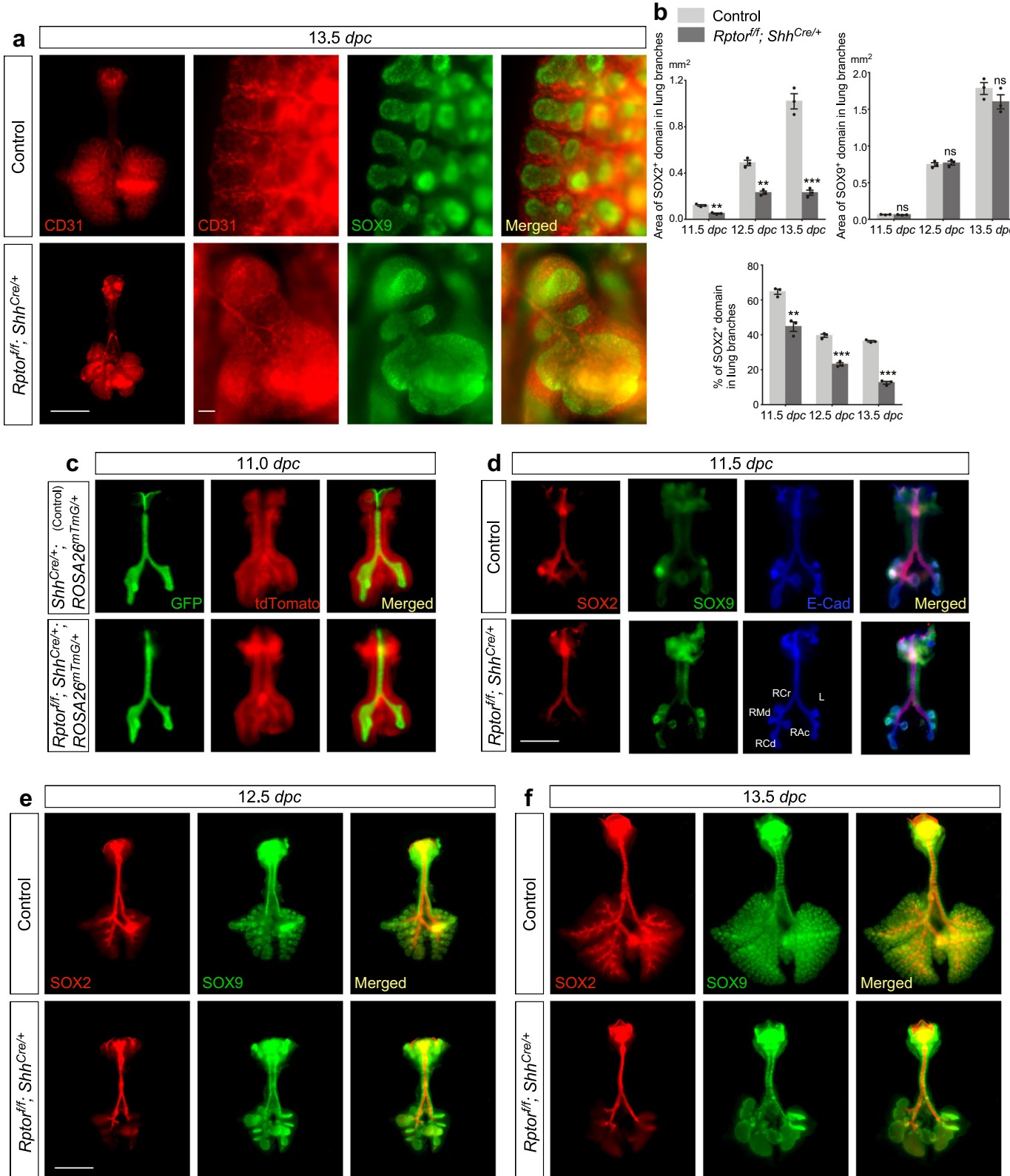

discerned at 13.5 *dpc* in *Tfam^f/f; Shh^Cre/+* lungs and 12.75 *dpc* in *Cox10^f/f; Shh^Cre/+* lungs (Fig. 6i). Nevertheless, the main characteristics of the branching phenotype were similar among the three mutants. The SOX2⁺ domain was also restricted to the trachea and main stem bronchi and some of the early branches in *Tfam^f/f; Shh^Cre/+* and *Cox10^f/f; Shh^Cre/+* lungs (Fig. 6i). Cell proliferation in the

lung epithelium was unaffected in the absence of *Tfam* or *Cox10* (Supplementary Fig. 10c). Thus, defective branching from SOX9⁺ cells failed to produce the conducting airways and led to cyst formation in *Tfam^f/f; Shh^Cre/+* and *Cox10^f/f; Shh^Cre/+* lungs.

We then evaluated saccule formation and differentiation of AT1 and AT2 cells in the SOX9⁺ domain of *Tfam^f/f; Shh^Cre/+* and *Cox10^f/f;*

**Fig. 2 | SOX2–SOX9 distribution is perturbed in *Rptor*-deficient lungs. a** Whole-mount immunostaining of dissected lungs from *Rptor^{f/f}; Shh^{Cre/+}* mice and control littermates at 13.5 *days post coitus (dpc)*. CD31 (PECAM1) labeled endothelial cells while SOX9 marked the distal airway epithelium. Scale bar = 1 mm for whole-lung images; scale bar = 100 μm for images with higher magnification. **b** Quantification of the area of SOX2⁺ and SOX9⁺ domains and the percentage of SOX2⁺ domain in lung branches in control and *Rptor^{f/f}; Shh^{Cre/+}* lungs from 11.5 to 13.5 *dpc* (*n* = 3 for each group). In the absence of epithelial *Rptor*, the area of the SOX9⁺ domain in the distal lung increased at a rate similar to that in control lungs. By contrast, the area of the SOX2⁺ domain failed to expand in the mutant lungs. This suggests that the SOX9 to SOX2 transition was blocked in *Rptor^{f/f}; Shh^{Cre/+}* lungs. **c** Whole-lung imaging of dissected lungs from *Shh^{Cre/+}; ROSA26^{mTmG/+}* (control) and *Rptor^{f/f}; Shh^{Cre/+};*

*ROSA26^{mTmG/+}* mice at 11.0 *days post coitus (dpc)*. GFP was activated from the *ROSA26^{mTmG}* allele in all lung epithelial cells by *Shh^{Cre}*; tdTomato marked all non-epithelial cells. Scale bar = 0.5 mm. **d** Whole-mount immunostaining of dissected lungs from *Shh^{Cre/+}; ROSA26^{mTmG/+}* and *Rptor^{f/f}; Shh^{Cre/+}; ROSA26^{mTmG/+}* mice at 11.5 *dpc*. E-Cadherin (E-Cad) labeled all epithelial cells. SOX9 labeled the distal airway epithelium, while SOX2 labeled the proximal airway epithelium. Scale bar = 0.5 mm. **e** Whole-mount immunostaining of dissected lungs from *Rptor^{f/f}; Shh^{Cre/+}* mice and control littermates at 12.5 *dpc*. Scale bar = 1 mm. **f** Whole-mount immunostaining of dissected lungs from *Rptor^{f/f}; Shh^{Cre/+}* mice and control littermates at 13.5 *dpc*. Scale bar = 1 mm. All values are mean ± SEM. (**) *p* < 0.01; (***) *p* < 0.001; ns not significant (two-tailed, unpaired Student's *t*-test). Source data are provided as a Source Data file.

*Shh^{Cre/+}* lungs. Saccule-like structures formed at the edge of the cysts in *Tfam*- and *Cox10*-deficient lungs at 16.5 *dpc* (Fig. 7a, c) and progressed to more elaborate structures at 18.5 *dpc* (Fig. 7b, d). Thus, reduced mitochondrial capacity disrupts branching morphogenesis to generate the conducting airways but preserves saccule formation.

The saccule-like structures in *Tfam^{f/f}; Shh^{Cre/+}* and *Cox10^{f/f}; Shh^{Cre/+}* lungs expressed AT1 markers (*e.g.*, T1α) and contained SOX9/SPC double positive cells at the tips (Figs. 7c, d and 8b, d, e and Supplementary Fig. 11) despite loss of MTCO1 (Fig. 8a, c). These results suggest compromised ATP production in the mutant lungs blocked the generation of SOX2⁺ cells from distal SOX9⁺ progenitors but did not prevent them from producing saccules and AT1/2 cells.

As an alternative approach to perturb mitochondrial function, we inactivated the mitochondria-associated protein, LRPPRC (leucine-rich PPR-motif containing), in the lung epithelium using *Shh^{Cre}* to convert a floxed allele of *Lrpprc*, *Lrpprc^{f37}*, into a null allele. LRPPRC regulates multiple aspects of mitochondrial function. *Lrpprc^{f/f}; Shh^{Cre/+}* mice displayed branching defects (Supplementary Figs. 12 and 13), which could be first detected at 15.5 *dpc* and were milder than those in *Tfam^{f/f}; Shh^{Cre/+}* and *Cox10^{f/f}; Shh^{Cre/+}* lungs. Lung cysts were observed mainly in the more distal part of *Lrpprc*-deficient lungs (Supplementary Figs. 12 and 13). Likewise, saccule-like structures that contained AT1 and AT2 cells developed along the edge of cysts in *Lrpprc^{f/f}; Shh^{Cre/+}* lungs (Supplementary Figs. 12 and 13).

Taken together, our findings further support the notion that saccule formation and AT1/2 differentiation can occur independently of lung branching and highlight a differential energy requirement for distinct cellular processes (Fig. 8f).

**The Hedgehog pathway acts downstream of the mTORC1-mitochondria axis in regulating formation of the conducting airways**

To explore the downstream targets of the mTORC1-mitochondria axis during formation of the conducting airways we performed RNA-Seq analysis of control and *Rptor^{f/f}; Shh^{Cre/+}*, *Tfam^{f/f}; Shh^{Cre/+}* and *Cox10^{f/f}; Shh^{Cre/+}* lungs at 14.5 *dpc* (Fig. 9a, b, c). In all three mutant lungs, *Sox2* expression levels were greatly reduced while *Sox9* expression levels were unchanged, suggesting a persistent failure of generating SOX2⁺ cells from SOX9⁺ progenitors. Moreover, expression levels of *Foxj1* (ciliated cells) and *Ascl1* (neuroendocrine cells) were reduced while expression levels of *Trp63* (p63) (basal cells) in the trachea and main stem bronchi were unaltered. The changes in molecular signature further confirmed a selective loss of the conducting airways through phenotypic analysis when the mTORC1-mitochondria axis is disrupted.

We found that the Hedgehog (Hh) signaling pathway was downregulated in the lungs of all three mutants (Fig. 9a, b, c). Loss of *Sonic hedgehog* (*Shh*) in the lung[38] generated phenotypes (Fig. 9d, e) similar to those in *Rptor^{f/f}; Shh^{Cre/+}*, *Tfam^{f/f}; Shh^{Cre/+}* and *Cox10^{f/f}; Shh^{Cre/+}* lungs. The conducting airways located between the main stem bronchi (p63⁺)

and saccules (SPC⁺HOPX⁺) failed to form in *Shh^{−/−}* lungs (Fig. 9f, g). Moreover, the cystic surface in *Shh* mutant lungs was populated by SOX9⁺ patches (Supplementary Fig. 14). These results suggest that the mTORC1-mitochondria axis controls a signaling network that includes the Hh pathway.

## Discussion

Our work illustrates how distinct cellular processes exhibit differential energy dependence. Using lung development as a model system, we have provided insight into how different stages of lung development are controlled. In particular, lung branching and saccule formation can be independently regulated. These findings serve as the paradigm for examining other cellular processes during tissue and organ formation.

Proximal-distal patterning of lung branches by expansion of SOX9⁺ progenitors and production of SOX2⁺ cells from SOX9⁺ progenitors has been well described[10]. However, the molecular basis of this process remains elusive. Through manipulation of mTORC1 signaling and mitochondrial capacity, we were able to dissect distinct steps of SOX9⁺ cell expansion, SOX9–SOX2 transition to form the conducting airways, and differentiation of SOX9⁺ cells to generate alveolar epithelial cells and form saccules (Fig. 8f). In the absence of *Rptor*, *Tfam* or *Cox10* in the lung epithelium, SOX9–SOX2 transition is blocked and the conducting airways fail to form. However, SOX9⁺ cell expansion and saccule formation are unaffected (Fig. 8f). Our results suggest that formation of the conducting airways and saccules can be regulated independently.

Our analysis of *Rptor*, *Tfam* or *Cox10* mutant lungs reveals that the distal SOX9⁺ progenitor pool is preserved during branching morphogenesis despite a failure to produce SOX2⁺-conducting airways (Figs. 8f and 10). During sacculation of the mutant lungs, HOPX⁺ AT1 cells are adjacent to the SOX2⁺ trachea and main stem bronchi. By contrast, SOX9⁺ AT2 cells are located primarily at the tips of saccules. This suggests that the distal SOX9⁺ progenitors in control lungs proliferate to maintain a pool of progenitors at the tip of the lung buds (tip progenitors). Tip progenitors differentiate to precursors of AT1 cells (transition progenitors), which are located proximally to the progenitor pool at the tip. Transition progenitors would then form SOX2⁺ progenitors while downregulating SOX9 expression to construct the conducting airways. In *Rptor*, *Tfam* or *Cox10* mutant lungs, the tip progenitors can still expand and produce transition progenitors. However, derivation of SOX2⁺ cells from transition progenitors is blocked in the mutant lungs, resulting in loss of the conducting airways.

Formation of saccules on the cystic surface of mTORC1-deficient lungs or from the cystic edge of *Tfam*- or *Cox10*-deficient lungs facilitates visualization of saccule formation to further test our model. We were able to examine the rudimentary saccule structure at 15.5 *dpc*, the progression of saccule development at 16.5 *dpc* and the maturation of saccules at 18.5 *dpc* in the mutant lungs. Rudimentary saccule progenitors consist of SOX9⁺ cells (tip

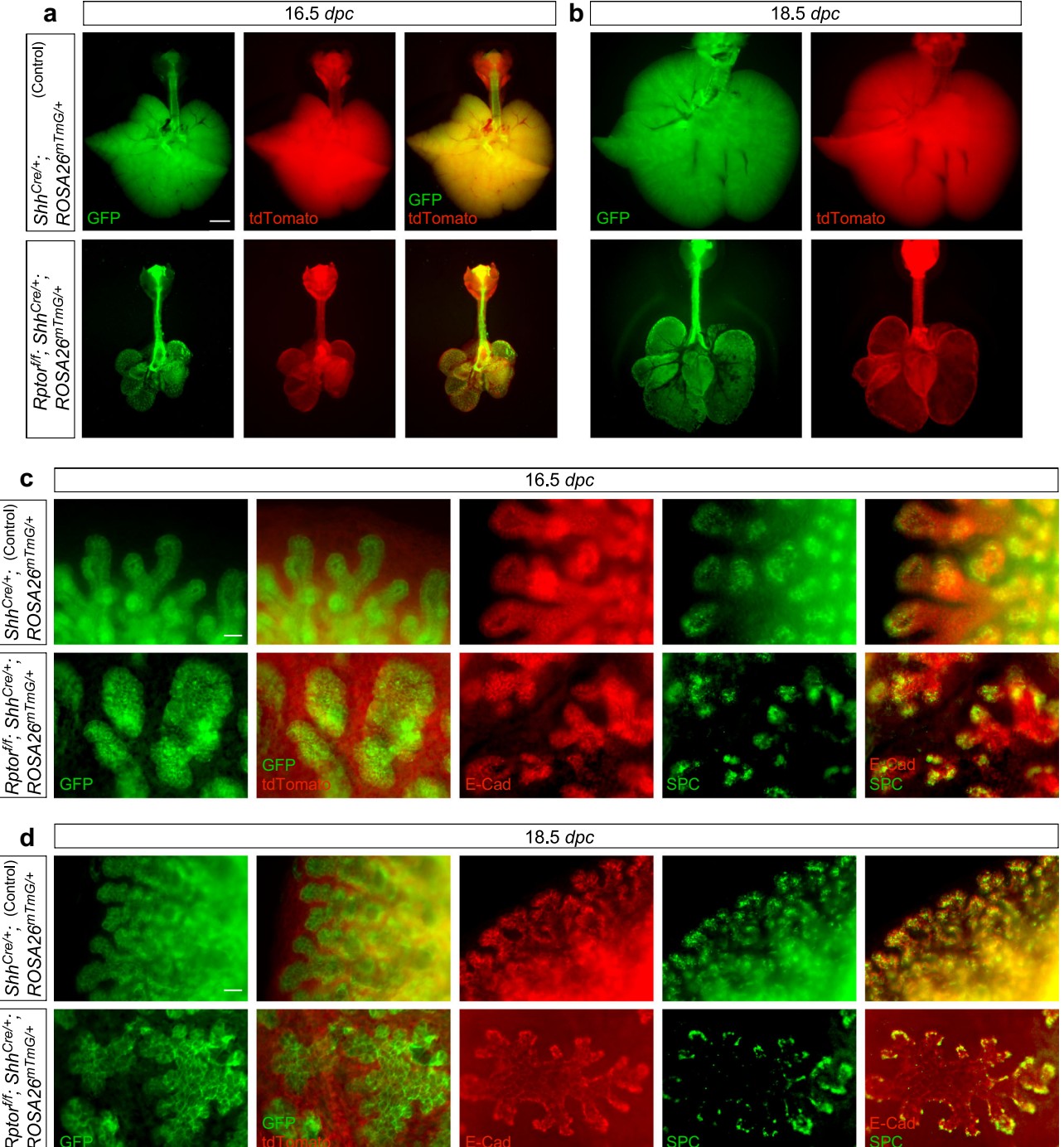

**Fig. 3 | Loss of *Rptor* does not disrupt saccule formation. a, b** Whole-lung imaging of dissected lungs from *Shh^{Cre/+}; ROSA26^{mTmG/+}* (control) and *Rptor^{f/f}; Shh^{Cre/+}; ROSA26^{mTmG/+}* mice at 16.5 and 18.5 *days post coitus (dpc)*. GFP was activated from the *ROSA26^{mTmG}* allele in all lung epithelial cells by *Shh^{Cre}*; tdTomato marked all non-epithelial cells. Many GFP⁺ dots were present on the surface of *Rptor^{f/f}; Shh^{Cre/+}; ROSA26^{mTmG/+}* lungs. Scale bar = 1 mm. **c** Higher magnification of whole-lung imaging and whole-mount immunostaining of dissected lungs from *Shh^{Cre/+};ROSA26^{mTmG/+}* and *Rptor^{f/f}; Shh^{Cre/+};ROSA26^{mTmG/+}* mice at 16.5 *dpc*. GFP and E-Cadherin (E-Cad) labeled all epithelial cells, while SPC (SFTPC) marked alveolar epithelial type II (AT2) cells located at the tips of saccules. Many GFP⁺ dots and patches were found on the surface of *Rptor^{f/f}; Shh^{Cre/+}; ROSA26^{mTmG/+}* lungs at 16.5 *dpc*. SPC distribution in saccule-like structures in *Rptor^{f/f}; Shh^{Cre/+}; ROSA26^{mTmG/+}* lungs was similar to that in control lungs at 16.5 *dpc*. Scale bar = 25 μm. **d** Higher magnification of whole-lung imaging and whole-mount immunostaining of dissected lungs from *Shh^{Cre/+};ROSA26^{mTmG/+}* and *Rptor^{f/f}; Shh^{Cre/+}; ROSA26^{mTmG/+}* mice at 18.5 *dpc*. GFP and E-Cadherin (E-Cad) labeled all epithelial cells, while SPC marked alveolar epithelial type II (AT2) cells located at the tip of saccules. Developmental progression of SPC distribution from 16.5 to 18.5 *dpc* followed a similar pattern between control and *Rptor*-deficient lungs. Scale bar = 25 μm.

progenitors). During saccule development, SOX9⁺ cells are confined to the tips of saccules and express SPC. By contrast, SOX9⁻ cells within the saccules are associated with HOPX expression. When saccules mature, the SOX9 signal sharply decreases; the tips of saccules are occupied by SPC⁺ AT2 cells while the bulk of saccules is composed of HOPX⁺ AT1 cells. This is consistent with the notion that the tip progenitors produce SPC⁺ AT2 cells while the transition progenitors fail to generate SOX2⁺ cells and become

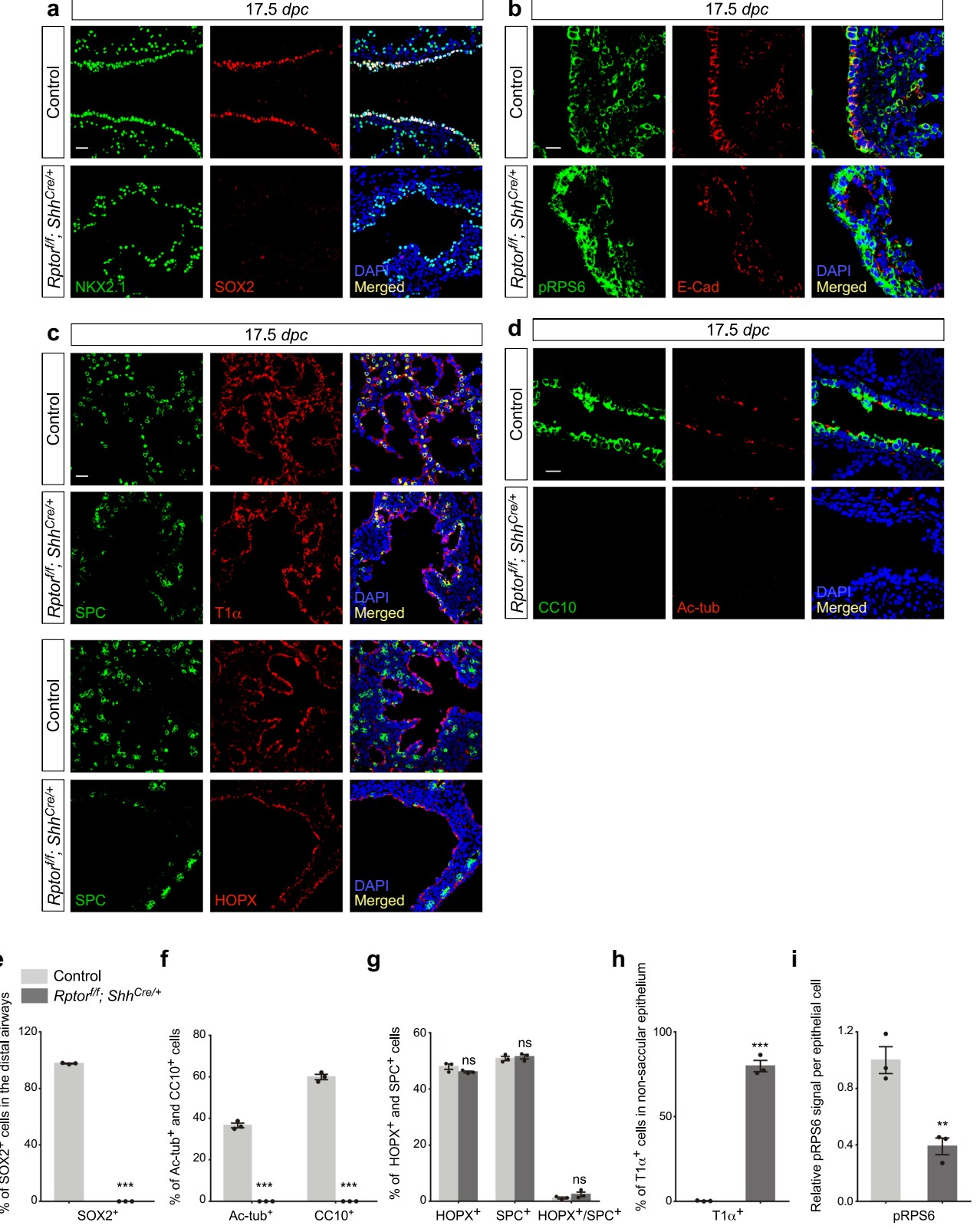

HOPX⁺ AT1 cells in the mutant lungs. This sequence of saccule formation is consistent with the published data using lineage tracing and scRNA-Seq[39]. Our work provides a framework for further studies to trace saccule formation dynamically by combining live imaging, organ culture and molecular tools.

The main characteristics of the lung phenotypes are similar among the *Rptor*, *Tfam* or *Cox10* mutants in this study. Nevertheless,

their defects are not identical. We surmise that this is due to the fact that mTORC1 controls multiple cellular processes in addition to mitochondrial capacity. We noticed that the majority of the newly generated epithelial cells in *Rptor^{f/f}; Shh^{Cre/+}* lungs would fill the inside of the distal end of lung branches, forming solid sphere-like structures. The distal SOX9⁺ progenitors sit on the surface of these structures. By contrast, newly produced epithelial cells in *Tfam^{f/f};*

**Fig. 4 | Cell type specification occurs in *Rptor*-deficient saccules despite loss of cell types in the conducting airways. a** Immunostaining of lung sections collected from *Rptor^f/f; Shh^Cre/+* mice and control littermates at 17.5 days post coitus (dpc). NKX2.1 labeled all epithelial cells, while SOX2 labeled the conducting airways. Scale bar = 25 μm. **b** Immunostaining of lung sections collected from *Rptor^f/f; Shh^Cre/+* mice and control littermates at 17.5 dpc. E-Cadherin (E-Cad) labeled all epithelial cells including the epithelium of the airways and saccules; pRPS6 is a marker of mTORC1 activity. pRPS6 signal was sharply reduced in the lung epithelium of *Rptor^f/f; Shh^Cre/+* mice at 17.5 dpc. Scale bar = 10 μm. **c** Immunostaining of lung sections collected from *Rptor^f/f; Shh^Cre/+* mice and control littermates at 17.5 dpc. SPC labeled alveolar epithelial type II (AT2) cells, while T1α (PDPN) or HOPX marked alveolar epithelial type I (AT1) cells. Scale bar = 25 μm. **d** Immunostaining of lung sections collected from *Rptor^f/f; Shh^Cre/+* mice and control littermates at 17.5 dpc. CC10 marked club

(Clara) cells and Ac-tub marked ciliated cells in the airways. Scale bar = 10 μm. **e** Quantification of the percentage of SOX2⁺ cells in the conducting airways (n = 3 for each group), shown in (**a**). **f** Quantification of the percentage of Ac-tub⁺ and CC10⁺ cells in the conducting airways (n = 3 for each group), shown in (**d**). **g** Quantification of the percentage of HOPX⁺ (AT1) cells and SPC⁺ (AT2) cells in the saccules (n = 3 for each group), shown in (**c**). **h** Quantification of the percentage of T1α⁺ cells in the non-saccular epithelium (excluding the trachea and main stem bronchi) (n = 3 for each group), shown in (**c**). **i** Quantification of the relative pRPS6 signal in the lung epithelium (n = 3 for each group), shown in (**b**). Five views per lung (a total of 15 views) were used for counting in (**e–i**). All values are mean ± SEM. (**) $p < 0.01$; (***) $p < 0.001$; ns not significant (two-tailed, unpaired Student's t-test). Source data are provided as a Source Data file.

*Shh^Cre/+* and *Cox10^f/f; Shh^Cre/+* lungs would push outward, resulting in cyst formation. The distal SOX9⁺ progenitors settle to the edge of the cysts. Thus, in *Rptor^f/f; Shh^Cre/+* lungs, saccules sprout from the surface of the defective lobules while saccules in *Tfam^f/f; Shh^Cre/+* and *Cox10^f/f; Shh^Cre/+* lungs emerge from the edges of the cystic lobules. These results form the basis of future investigation to study the molecular basis of lung branching and saccule formation.

Emergence of saccules in the absence of lung branching in *Rptor*, *Tfam*, *Cox10* or *Shh* mutant lungs suggests that these two processes can proceed independently (Fig. 10). While saccule formation normally commences once lung branching is completed, saccule formation can occur at the proper developmental time when lung branching is blocked. This infers that a program for saccule formation can be activated without the completion of lung branching and without an existing scaffold. It also indicates the presence of a developmental clock. In *Sox9* mutant lungs, certain marker genes for differentiated alveolar cells were upregulated or prematurely expressed[40,41]. This suggests that SOX9 suppresses premature initiation of alveolar differentiation. It is possible that SOX9 is involved in controlling the timing of alveolar differentiation. Our work has set the stage for identifying the regulatory programs that execute distinct steps of lung development and the developmental clock that controls lung development.

Transcriptome analysis and genetic studies suggest that Hh signaling functions as a downstream effector of the mTORC1-mitochondria axis. Loss of the conducting airways while preserving sacculation are common features in *Rptor*, *Tfam*, *Cox10* and *Shh* mutants. Nevertheless, the size of the lung and cysts are smaller in *Shh* mutant lungs. We speculate that Hh signaling has additional functions in addition to mediating the effects of the mTORC1-mitochondria axis. Moreover, KEGG pathway analysis revealed several pathways perturbed in *Rptor*, *Tfam* and *Cox10* mutant lungs (Supplementary Fig. 15), including the Rap1 signaling pathway, the PI3K-Akt signaling pathway and the Ras signaling pathway. The mitochondrial pathway was not among the top 40 enriched pathways (Supplementary Fig. 15). This is likely because *Rptor* controls mitochondrial genes posttranscriptionally. We noted that Kras activation in the lung epithelium was reported to disrupt SOX9 to SOX2 transition[40], a common defect shared by *Rptor*-, *Tfam*- and *Cox10*-deficient lungs. Taken together, we propose that mTORC1 signaling functions in a signaling network, part of which involves mitochondrial capacity, and Hh and Kras signaling. These observations provide molecular insight into how mTORC1 signaling, mitochondrial capacity and the downstream signaling events control branching morphogenesis. Nevertheless, the detailed molecular mechanisms by which mitochondrial capacity and ATP production regulate branching require additional investigation.

mTORC1 regulates multiple cellular events including mitochondrial function, metabolism, protein synthesis and autophagy. Several targets (e.g., eukaryotic translation initiation factor 4E-binding protein

(4E-BP), ribosomal protein S6 kinase (S6K) and unc-51 like kinase (ULK)) of mTORC1 have been identified. It is interesting to note that a connection established in cell lines is not necessarily recapitulated in the animals. For instance, disruption of *4e-bp1*, *4e-bp2*, *S6K1* and *Ulk1/2* in mice does not lead to severe branching defects in the lungs as reported in the literature[42–44]. Thus, genetic studies are required to identify the functional targets of mTORC1 in vivo. Our results in this study suggest that mTORC1 controls mitochondrial capacity during lung branching. mTORC1 is a repressor of mTORC2 signaling[45]. This raises the possibility that the phenotypes in mTORC1 mutant lungs were caused by mTORC2 activation. However, it was reported that active mTOR signaling in the lung does not lead to branching defects[46]. Thus, it is unlikely that mTORC2 activation underlies the defects in mTORC1-deficient lungs. In summary, this work has advanced our molecular understanding of lung branching and sacculation and provides the infrastructure for future studies.

## Methods

### Animal husbandry

The mouse experiments in this study were performed following the protocols (AN187712) approved by the Institutional Animal Care and Use Committee (IACUC) of the University of California, San Francisco (UCSF). Mice were housed in a SPF (specific pathogen free) facility with controlled temperature and humidity and a 12-hour dark/12-hour light cycle. Details on the mouse ages and genotypes are included in the figures and main text. The mouse lines used are listed below. *Rptor^f* [B6.Cg-*Rptor^tm1.1Dmsa*/J], *Tfam^f* [B6.Cg-*Tfam^tm1.1Ncdl*/J], *Cox10^f* [B6.129×1-*Cox10^tm1Ctm*/J], *Lrpprc^f* [*Lrpprc^tm1.1Lrsn*/J], *ROSA26^mTmG* [*Gt(ROSA)26Sor^tm4(ACTB-tdTomato,-EGFP)Luo*/J], *ROSA26^tdTomato* [B6;129S6-*Gt(ROSA)26Sor^tm14(CAG-tdTomato)Hze*/J] and *Shh^Cre* [B6.Cg-*Shh^tm1(EGFP/cre)Cjt*/J] were obtained from Jackson Laboratory (Bar Harbor, ME, USA).

### Histology and immunofluorescence

Histology and immunofluorescence were performed as previously described[31,47]. In brief, mouse lungs at the indicated time points were dissected and fixed with 4% paraformaldehyde (PFA) (MilliporeSigma, Cat# P6148) on ice for 1 h. The lungs were embedded either in OCT or paraffin wax and sectioned at 7 μm.

For immunofluorescence, the primary antibodies used were: chicken anti-GFP (1:200, abcam, Cat# ab13970; RRID:AB_300798), rabbit anti-NKX2.1 (1:100, Epitomics, Cat# 2044–1; RRID:AB_1267367), goat-anti-CC10 (1:200, Santa Cruz Biotechnology, Cat# sc-9773; RRID:AB_2183391), mouse anti-acetylated tubulin (1:200, MilliporeSigma, Cat# T6793; RRID:AB_477585), rabbit anti-prosurfactant protein C (proSP-C) (1:200, MilliporeSigma, Cat# AB3786; RRID:AB_91588), hamster anti-T1α (1:200, Developmental Studies Hybridoma Bank, Cat# 8.1.1; RRID:AB_531893), mouse anti-HOPX (1:100, Santa Cruz Biotechnology, Cat# sc-398703; RRID:AB_2687966), mouse anti-p63 (1:100, Santa Cruz Biotechnology, Cat# sc-8431; RRID:AB_628091), rabbit anti-MPC1 (1:100, MilliporeSigma, Cat# HPA045119; RRID:AB_10960421), rat anti-E cadherin (1:200, Life Technologies, Cat# 13-

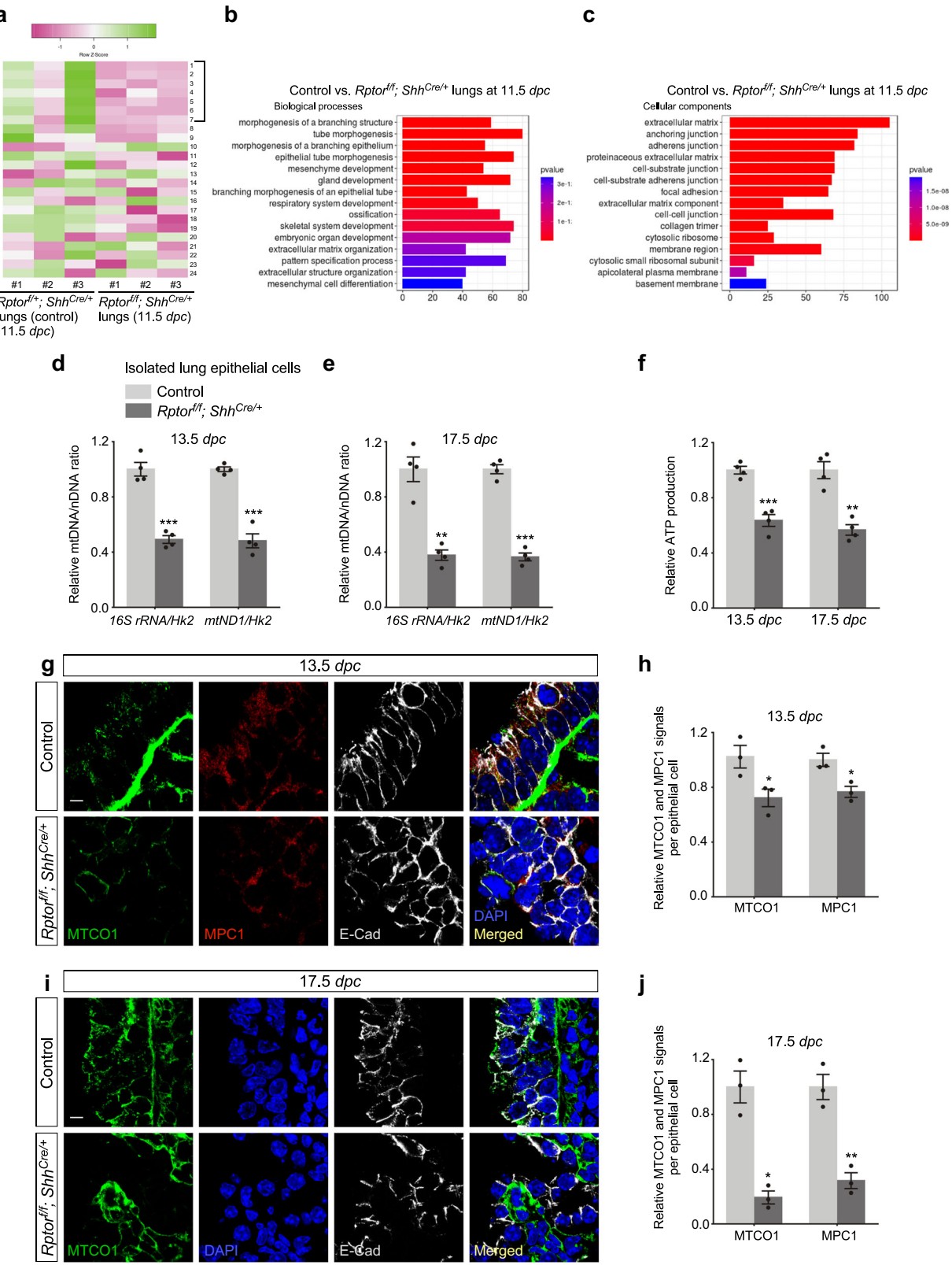

1900; RRID:AB_2533005), mouse anti-β-catenin (1:100, BD Transduction Laboratories, Cat# 610154; RRID:AB_397555), mouse anti-MTCO1 (1:100, abcam, Cat# ab14705; RRID:AB_2084810), mouse anti-GM130 (1:100, BD Biosciences, Cat# 610822; RRID:AB_398141), mouse anti-ACTA2 (1:200, Thermo Scientific Lab Vision, Cat# MS-113-P0; RRID:AB_64001), rat anti-PECAM-1 (CD31) (1:150, Santa Cruz Biotechnology, Cat# sc-18916; RRID:AB_627028), rabbit anti-PDGFRA (1:150, Cell Signaling Technology, Cat# 3164; RRID:AB_2162351), mouse anti-S6 Ribosomal Protein (1:100, Cell Signaling Technology, Cat# 2317; RRID:AB_2238583), rabbit anti-Phospho-S6 Ribosomal Protein (Ser235/236) (1:100, Cell Signaling Technology, Cat# 4856; RRID:AB_2181037), rabbit anti-SOX2 (D9B8N) (1:200, Cell Signaling

**Fig. 5 | Mitochondrial capacity is reduced in *Rptor*-deficient lungs. a** Heatmap of gene expression of the glycolytic and related pathways from control and *Rptor^{f/f}; Shh^{Cre/+}* lungs at 11.5 *days post coitus* (*dpc*). The genes with significantly altered expression in the absence of *Rptor* were: *Ldha* (1), *Aldoa* (2), *Pgk1* (3), *Pfkl* (*Pfkm*) (4), *Tpi1* (5), *Hk2* (6) and *Pklr* (7). The expression of these seven genes was decreased in *Rptor*-deficient lungs. **b**, **c** Gene Ontology (GO) pathway analysis of bulk RNA-Seq of control and *Rptor^{f/f}; Shh^{Cre/+}* lungs at 11.5 *dpc*. **d** Quantification of the relative ratio of mitochondrial DNA (mtDNA) to nuclear DNA (nDNA) in purified lung epithelial cells derived from control and *Rptor^{f/f}; Shh^{Cre/+}* mice at 13.5 *dpc* (*n* = 4 for each group). *16S rRNA* and *mtND1* represent the mitochondrial genes while *Hk2* (hexokinase 2) represents the nuclear genes. **e** Quantification of the relative ratio of mtDNA to nDNA in purified lung epithelial cells derived from control and *Rptor^{f/f}; Shh^{Cre/+}* mice at 17.5 *dpc* (*n* = 4 for each group). **f** Measurement of relative ATP production in purified lung epithelial cells derived from control and *Rptor^{f/f}; Shh^{Cre/+}* mice at 13.5 (*n* = 4 for each group) and 17.5 (*n* = 4 for each group) *dpc*. **g** Immunostaining of lung sections collected from *Rptor^{f/f}; Shh^{Cre/+}* mice and control littermates at 13.5 *dpc*. E-Cadherin (E-Cad) labeled all epithelial cells. The MTCO1 and MPC1 signals indicate the mitochondrial mass within individual epithelial cells. A reduction of MTCO1 and MPC1 signal in the lung epithelium of *Rptor^{f/f}; Shh^{Cre/+}* mice was detected. Scale bar = 5 µm. **h** Quantification of the relative MTCO1 and MPC1 signals in the lung epithelium (*n* = 3 for each group). Five views per lung (a total of 15 views) were used for counting. The signal density in control and mutant lungs was measured using ImageJ, and then divided by the corresponding number of cells. **i** Immunostaining of lung sections collected from *Rptor^{f/f}; Shh^{Cre/+}* mice and control littermates at 17.5 *dpc*. MTCO1 signal was decreased in the lung epithelium of *Rptor^{f/f}; Shh^{Cre/+}* mice. Scale bar = 5 µm. **j** Quantification of the relative MTCO1 and MPC1 signals in the lung epithelium (*n* = 3 for each group). Five views per lung (a total of 15 views) were used for counting as in (**h**). All values are mean ± SEM. (*) $p < 0.05$; (**) $p < 0.01$; (***) $p < 0.001$; ns not significant (two-tailed, unpaired Student's *t*-test). Source data are provided as a Source Data file.

Technology, Cat# 23064; RRID:AB_2714146), goat anti-SOX9 (1:200, R&D Systems, Cat# AF3075; RRID:AB_2194160), rabbit anti-pMLC (S19) (1:100, Cell Signaling Technology, Cat# 3671; RRID:AB_330248), rabbit anti-PKC Zeta (C-20) (1:100, Santa Cruz Biotechnology, Cat# sc-216; RRID:AB_2300359). Secondary antibodies and conjugates used were: Alexa Fluor® 488 donkey anti-chicken antibody (1:1000, Jackson ImmunoResearch Laboratories, Cat# 703-546-155; RRID:AB_2340376), Alexa Fluor® 488 donkey anti-goat (1:1000, Life Technologies, Cat# A11055; RRID:AB_2534102), Alexa Fluor® 488 donkey anti-mouse (1:1000, Life Technologies, Cat# A21202; RRID:AB_141607), Alexa Fluor® 594 donkey anti-mouse (1:1000, Life Technologies, Cat# A21203; RRID:AB_141633), Alexa Fluor® 647 donkey anti-mouse (1:1000, Life Technologies, Cat# A31571; RRID:AB_162542), Alexa Fluor® 488 donkey anti-rabbit (1:1000, Life Technologies, Cat# A21206; RRID:AB_2535792), Alexa Fluor® 594 donkey anti-rabbit (1:1000, Life Technologies, Cat# A21207; RRID:AB_141637), Alexa Fluor® 594 donkey anti-rat (1:1000, Life Technologies, Cat# A21209; RRID:AB_2535795), Biotinylated goat-anti hamster (1:1000, Vector Laboratories, Cat# BA-9100; RRID:AB_2336137), Biotin-SP-conjugated AffiniPure donkey anti-rabbit (1:1000, Jackson ImmunoResearch Laboratories, Cat# 711-065-152; RRID:AB_2340593), Biotin-SP-conjugated AffiniPure donkey anti-rat (1:1000, Jackson ImmunoResearch Laboratories, Cat# 712-065-150; RRID:AB_2340646), Biotinylated horse anti-mouse (1:1000, Vector Laboratories, Cat# BA-2000; RRID:AB_2313581), Streptavidin, Alexa Fluor® 488 conjugate antibody (1:1000, Life Technologies, Cat# S11223), Streptavidin, Alexa Fluor® 594 conjugate antibody (1:1000, Life Technologies, Cat# S11227), Streptavidin, Alexa Fluor® 647 conjugate antibody (1:1000, Jackson ImmunoResearch Laboratories, Cat# 016-600-084; RRID:AB_2341101). The signal was detected using streptavidin-conjugated Alexa Fluor 488, 594, or 647 (1:1000, Life Technologies) or HRP-conjugated streptavidin (1:1000, TSA kit; Perkin-Elmer, Cat# NEL753001KT) coupled with fluorogenic substrate Alexa Fluor 594 or 488 tyramide for 30 s (1:200, TSA kit; Perkin-Elmer, Cat# NEL753001KT). F-actin was stained with rhodamine-conjugated phalloidin (1:200; Thermo Fisher, Cat# R415) in PBS for 2 h. Confocal images were captured on a Leica SPE laser-scanning confocal microscope. Adjustment of red/green/blue/gray histograms and channel merges were performed using LAS AF Lite (Leica Microsystems).

### Whole-mount immunostaining
Whole-mount immunostaining was performed as previously described[48,49]. Briefly, the embryonic lungs at the indicated time points were collected and fixed in 4% PFA on ice for 1 h. Samples were washed with 0.1% Tween-20/PBS for 30 min then dehydrated in graded methanol solutions (25%, 50%, 75%, 100%). After incubating in 5% $H_2O_2$/methanol overnight, the lungs were rehydrated through graded methanol solutions (100%, 75%, 50%, 25%, 0%) which diluted in 0.1% Tween-20/PBS and incubated in blocking buffer (1.5% BSA/0.5% Triton X-100/PBS) for 6 h. Primary antibodies were incubated at 4 °C overnight. On the next day, lungs were washed with blocking buffer for 5 h, then incubated with secondary antibodies at 4 °C overnight. Images were captured on a Nikon Eclipse E1000 microscope with a SPOT 2.3 CCD camera. The primary antibodies used were: rat anti-E Cadherin (1:200, Life Technologies, Cat# 13-1900; RRID:AB_2533005), rabbit anti-SOX2 (D9B8N) (1:200, Cell Signaling Technology, Cat# 23064; RRID:AB_2714146), goat anti-SOX9 (1:200, R&D Systems, Cat# AF3075; RRID:AB_2194160), rabbit anti-prosurfactant protein C (proSP-C) (1:200, MilliporeSigma, Cat# AB3786; RRID:AB_91588), rat anti-PECAM-1 (CD31) (1:150, Santa Cruz Biotechnology, Cat# sc-18916; RRID:AB_627028).

### Cell proliferation assays
Proliferation of lung epithelial cells was evaluated by EdU incorporation assays as previously described[48]. Briefly, pregnant females at the indicated time points were intraperitoneally injected with 300 µl of 5 mg/ml EdU (Research Products International, Cat# B71800-1.0)/PBS solution 1 h before dissection. The Click-iT EdU Alexa Fluor 488 Imaging Kit (Thermo Fisher, Cat# C10337) was performed to assess EdU incorporation. To measure the rate of epithelial cell proliferation, EdU staining was co-stained with E-Cad, a pan-epithelial cell marker. Nuclear DAPI helped identify individual cells. The EdU signal is nuclear and co-localizes with the DAPI signal. The number of proliferating epithelial cells (EdU$^+$ E-Cad$^+$ DAPI$^+$) and the total number of epithelial cells (E-Cad$^+$ DAPI$^+$) were manually counted using ImageJ. The rate of epithelial cell proliferation was calculated as the ratio of the number of EdU$^+$ E-Cad$^+$ DAPI$^+$ cells to the number of E-Cad$^+$ DAPI$^+$ cells, which was displayed as (EdU$^+$ E-Cad$^+$)/(E-Cad$^+$) for simplicity in the figures or legends. For each animal, at least 5 sections were used for counting.

### Purification of lung epithelial cells
Lungs from *Rptor^{f/f}; Shh^{Cre/+}*, *Tfam^{f/f}; Shh^{Cre/+}* and *Cox10^{f/f}; Shh^{Cre/+}* mice and their littermate controls were dissected at the indicated time points, placed in the digestion solution (1.2 U/ml dispase (BD Biosciences, Cat# 354235), 0.5 mg/ml collagenase B (Roche, Cat# 11088815001) and 50 U/ml DNase I (QIAGEN, Cat# 79254)) and rocked at 37 °C for 1 h to release individual lungs cells. An equal volume of culture medium (DMEM (Mediatech, Cat# 10-013-CV) with 10% FBS (Gibco, Cat# 10437-028), 2x penicillin/streptomycin (Gibco, Cat# 15070-063) and 1x L-glutamine (Gibco, Cat# 25030-081)) was added to the digested lung samples, which were filtered through 40 µm cell strainers. After centrifugation at 600 × *g* for 10 min, the cell pellets were resuspended in 200 µl purification buffer (Phenol red-free DMEM with 0.2% BSA and 2% FBS). Biotin-conjugated EpCAM antibody (anti-CD326 (EpCAM), 1:100, eBioscience, Cat# 13-

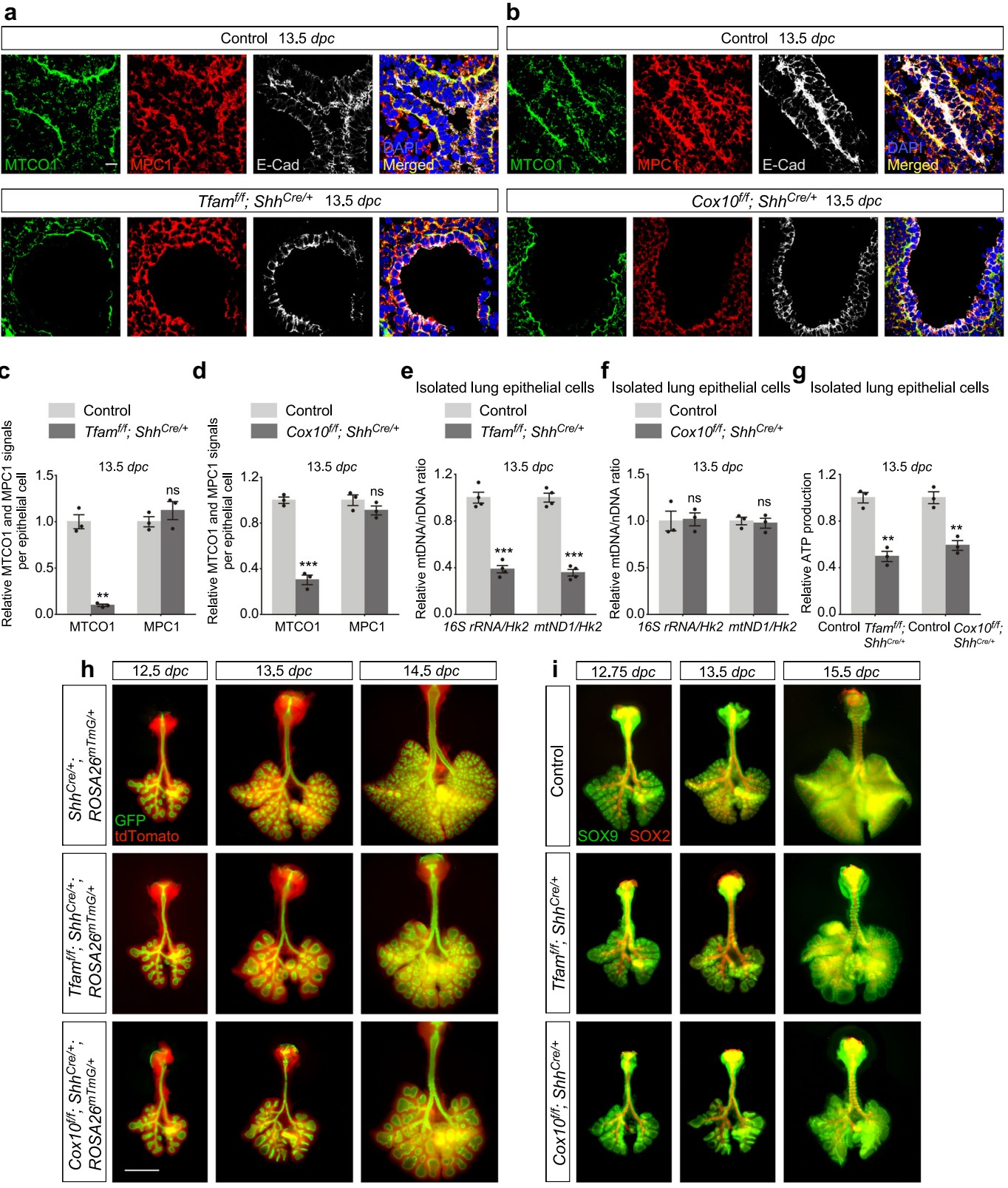

5791-82; RRID:AB_1659713) and Dynabeads M-280 Streptavidin (Invitrogen, Cat# 11205D) were added to the samples and incubated at 4 °C for 1 h with shaking. Epithelial cells were captured using Dynabeads magnets following the manufacturer's instructions. Purified lung epithelial cells were used for qPCR, ATP measurement and Western blotting analysis.

**Analysis of mtDNA/nDNA ratio**

mtDNA/nDNA ratio was evaluated as previously described[31]. Briefly, purified lung epithelial cells from *Rptor^{f/f}; Shh^{Cre/+}*, *Tfam^{f/f}; Shh^{Cre/+}*, *Cox10^{f/f}; Shh^{Cre/+}* mice and their littermate controls at 13.5 and 17.5 *dpc* were incubated in lysis buffer (100 mM Tris pH 7.5, 5 mM EDTA, 0.4% SDS, 200 mM NaCl, 50 µg /ml Proteinase K) at 55 °C overnight.

**Fig. 6 | Decreased mitochondrial function due to loss of *Tfam* or *Cox10* disrupts lung branching. a, b** Immunostaining of lung sections collected from control, *Tfam^{f/f}; Shh^{Cre/+}* and *Cox10^{f/f}; Shh^{Cre/+}* mice at 13.5 *days post coitum (dpc)*. E-Cadherin (E-Cad) labeled all epithelial cells. The MTCO1 and MPC1 signals indicate the mitochondrial mass within individual epithelial cells. A reduction in MTCO1 but not MPC1 signal in the lung epithelium of *Tfam^{f/f}; Shh^{Cre/+}* and *Cox10^{f/f}; Shh^{Cre/+}* mice was observed. Scale bar = 10 μm. **c, d** Quantification of the relative MTCO1 and MPC1 signals in the lung epithelium (*n* = 3 for each group). Five views per lung (a total of 15 views) were used for counting. The signal density in control and mutant lungs was measured using ImageJ, and then divided by the corresponding number of cells. **e, f** Quantification of the relative ratio of mitochondrial DNA (mtDNA) to nuclear DNA (nDNA) in purified lung epithelial cells derived from control, *Tfam^{f/f}; Shh^{Cre/+}* (*n* = 4 for each group) and *Cox10^{f/f}; Shh^{Cre/+}* (*n* = 3 for each group) mice at 13.5 *dpc*. *16S rRNA* and *mtND1* represent the mitochondrial genes while *Hk2* (hexokinase 2) represents the nuclear genes.

**g** Measurement of relative ATP production in purified lung epithelial cells derived from control, *Tfam^{f/f}; Shh^{Cre/+}* (*n* = 3 for each group) and *Cox10^{f/f}; Shh^{Cre/+}* (*n* = 3 for each group) mice at 13.5 *dpc*. **h** Whole-lung imaging of dissected lungs from *Shh^{Cre/+}; ROSA26^{mTmG/+}* (control), *Tfam^{f/f}; Shh^{Cre/+}; ROSA26^{mTmG/+}* and *Cox10^{f/f}; Shh^{Cre/+}; ROSA26^{mTmG/+}* mice at 12.5–14.5 *dpc*. GFP was activated from the *ROSA26^{mTmG}* allele in all lung epithelial cells by *Shh^{Cre}*; tdTomato marked all non-epithelial cells. Epithelial branching was perturbed in *Tfam-* and *Cox10*-deficient lungs. Scale bar = 1 mm. **i** Whole-mount immunostaining of dissected lungs from *Shh^{Cre/+}; ROSA26^{mTmG/+}* (control), *Tfam^{f/f}; Shh^{Cre/+}; ROSA26^{mTmG/+}* and *Cox10^{f/f}; Shh^{Cre/+}; ROSA26^{mTmG/+}* mice at 12.75, 13.5 and 15.5 *dpc*. SOX9 labeled the distal airway epithelium, while SOX2 marked the proximal airway epithelium. Perturbation of SOX9 to SOX2 differentiation could be detected as early as 12.75 *dpc* in *Cox10^{f/f}; Shh^{Cre/+}; ROSA26^{mTmG/+}* mice. Scale bar = 1 mm. All values are mean ± SEM. (**) *p* < 0.01; (***) *p* < 0.001; ns not significant (two-tailed, unpaired Student's *t*-test). Source data are provided as a Source Data file.

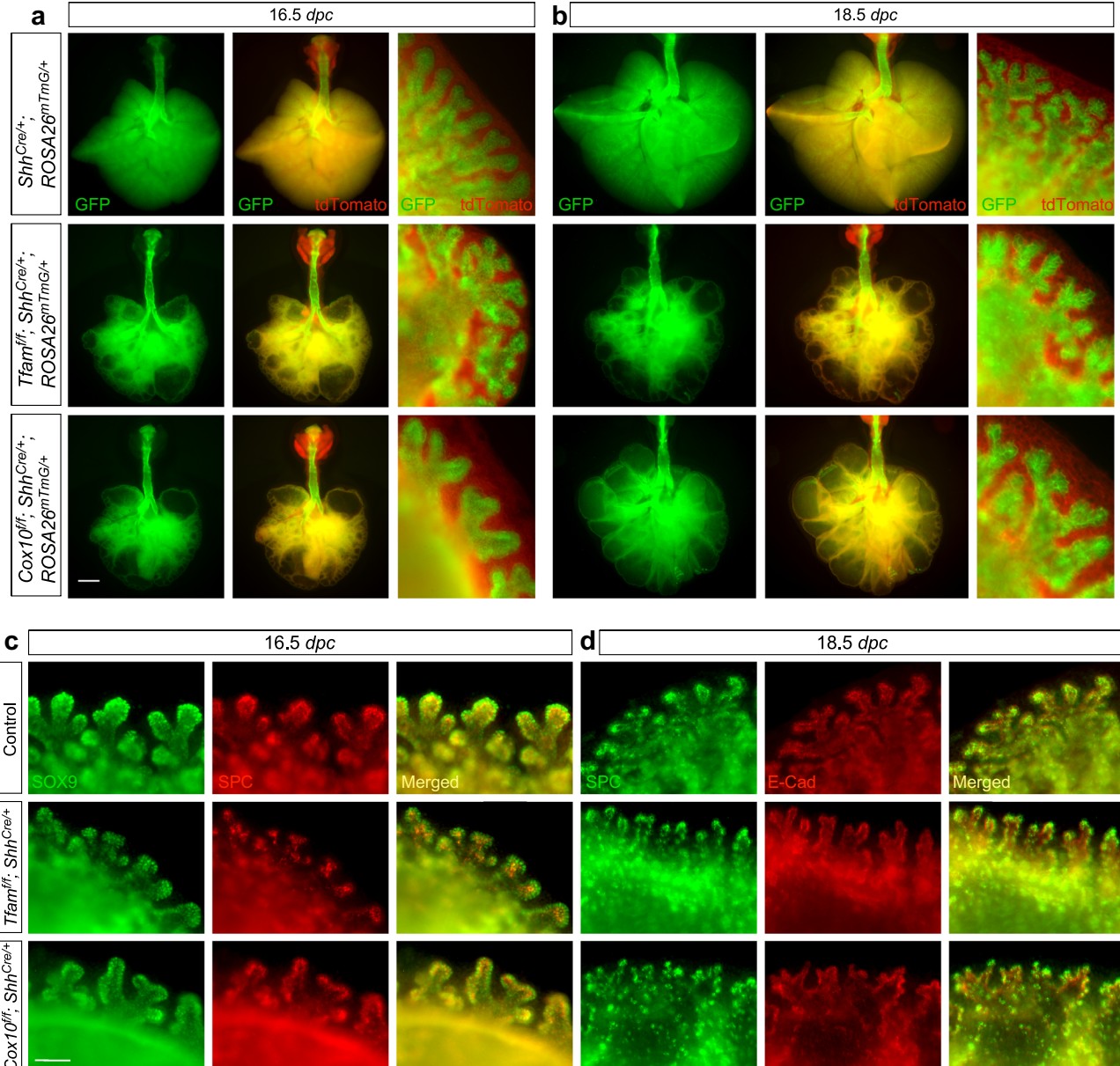

**Fig. 7 | Saccular formation progresses in the absence of *Tfam* or *Cox10*.**
**a, b** Whole-lung imaging of dissected lungs from *Shh^{Cre/+}; ROSA26^{mTmG/+}* (control), *Tfam^{f/f}; Shh^{Cre/+}; ROSA26^{mTmG/+}* and *Cox10^{f/f}; Shh^{Cre/+}; ROSA26^{mTmG/+}* mice at 16.5 and 18.5 *days post coitum (dpc)*. GFP was activated from the *ROSA26^{mTmG}* allele in all lung epithelial cells by *Shh^{Cre}*; tdTomato marked all non-epithelial cells. Many GFP+ tubules were present at the edge of the cystic regions of *Tfam-* or *Cox10*-deficient lungs at 16.5 *dpc*. These GFP+ patches would later develop into saccule-like structures at the saccular stage. Scale bar = 1 mm for whole-lung images; scale bar = 83 μm (see **c, d**)

for images with higher magnification. **c, d** Higher magnification of whole-mount immunostaining of dissected lungs from *Shh^{Cre/+}; ROSA26^{mTmG/+}* (control), *Tfam^{f/f}; Shh^{Cre/+}; ROSA26^{mTmG/+}* and *Cox10^{f/f}; Shh^{Cre/+}; ROSA26^{mTmG/+}* mice at 16.5 and 18.5 *dpc*. E-Cadherin (E-Cad) labeled all epithelial cells, while SPC marked alveolar epithelial type II (AT2) cells located at the tip of saccules. SOX9 and SPC distribution in saccule-like structures in *Tfam-* and *Cox10*-deficient lungs was similar to that in controls at 16.5 *dpc*. Development progression of SPC distribution from 16.5 to 18.5 *dpc* followed a similar pattern between control and mutant lungs. Scale bar = 83 μm.

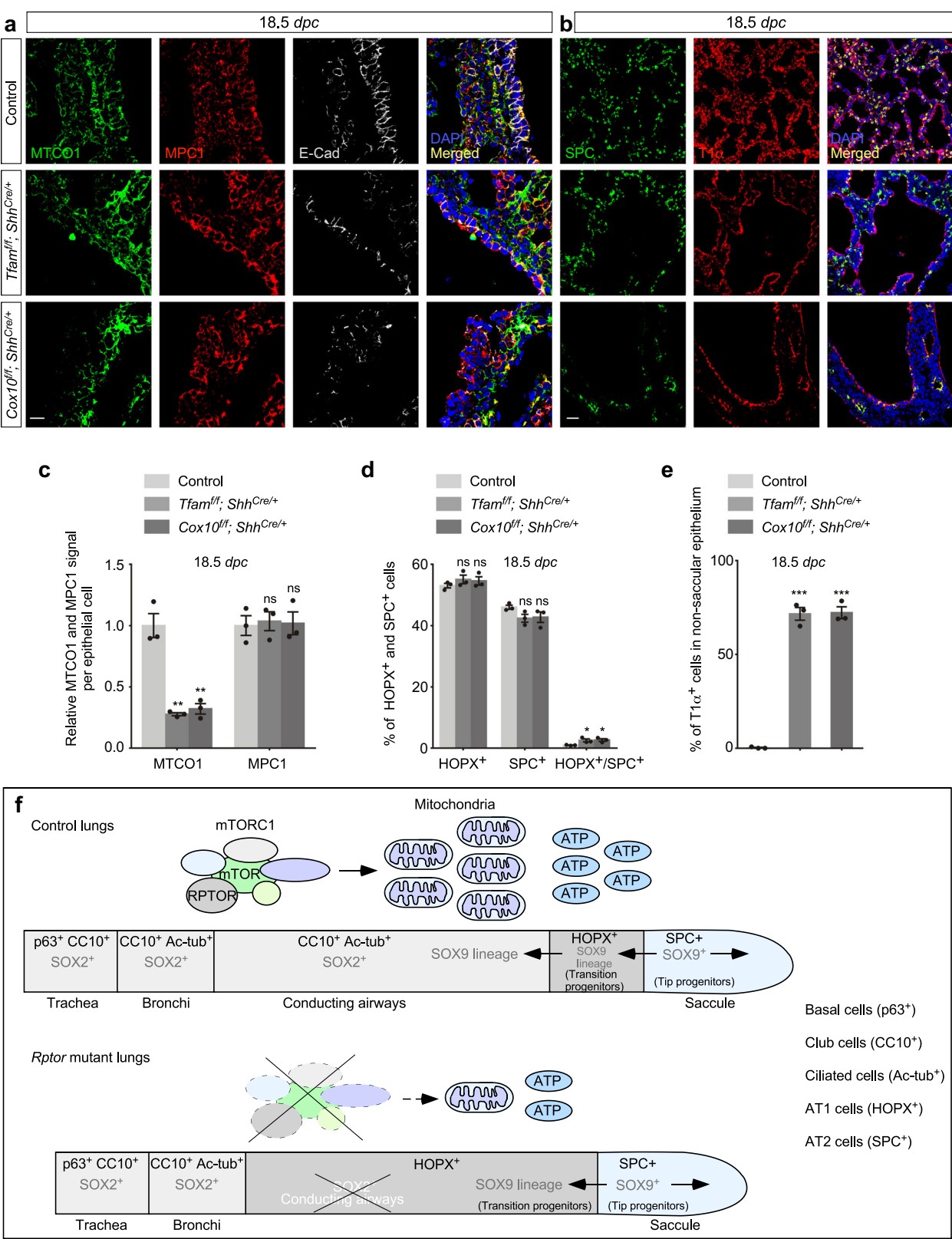

100 µg/ml RNase A was added and incubated at 37 °C for 30 min to degrade the RNAs. After mixing with an equal volume of phenol/chloroform, the samples were centrifuged at 13,400 × *g* for 10 min, and the lung DNA was concentrated by ethanol precipitation. Mitochondrial *16S rRNA* or *ND1* gene and the nuclear *Hk2* gene were used to assess the relative ratio of mitochondrial (mt) to nuclear (n) DNA copy number. Quantitative PCR was performed on an Applied Biosystems QuantStudio™ 5 Real-Time PCR System. The primer pairs used for the indicated genes were: *16S rRNA* forward, 5′-CCGCAAGGGAAA-GATGAAAG AC-3′; reverse, 5′-TCGTTTGGTTTCGGGGTTTC-3′, *mt-ND1* forward, 5′-CTAGCAGAAACAAACCGGGC-3′; reverse, 5′-CCGGCTGCGTATTCTAC

**Fig. 8 | Differentiation of AT1/2 cells within the saccules occurs in the absence of *Tfam* or *Cox10*. a** Immunostaining of lung sections collected from control, *Tfam^f/f; Shh^Cre/+* and *Cox10^f/f; Shh^Cre/+* mice at 18.5 *days post coitus (dpc)*. E-Cadherin (E-Cad) labeled all epithelial cells. The MTCO1 and MPC1 signals indicate the mitochondrial mass within individual epithelial cells. A reduction in MTCO1 but not MPC1 signal was detected in both *Tfam^f/f; Shh^Cre/+* and *Cox10^f/f; Shh^Cre/+* lungs. Scale bar = 10 μm. **b** Immunostaining of lung sections collected from control, *Tfam^f/f; Shh^Cre/+* and *Cox10^f/f; Shh^Cre/+* mice at 18.5 *dpc*. SPC labeled alveolar epithelial type II (AT2) cells, while T1α marked alveolar epithelial type I (AT1) cells. Scale bar = 25 μm. **c** Quantification of the relative MTCO1 and MPC1 signals in the lung epithelium (*n* = 3 for each group) at 18.5 *dpc*. Five views per lung (a total of 15 views) were used for counting. The signal density in control and mutant lungs was measured using ImageJ, and then divided by the corresponding number of cells. **d** Quantification of the percentage of HOPX^+ (AT1) cells and SPC^+ (AT2) cells in the saccules (*n* = 3 for each group). Five views per lung (a total of 15 views) were used for counting. **e** Quantification of the percentage of T1α^+ cells in the non-saccular epithelium (excluding the trachea and main stem bronchi) (*n* = 3 for each group). Five views per lung (a total of 15 views) were used for counting. **f** Schematic diagram of how the mTORC1-

mitochondria axis regulates lung branching morphogenesis. A pool of SOX9^+ progenitors at the distal epithelium (tip progenitors) expands to produce SOX9^+ transition progenitors, which generate SOX2^+ progeny in the conducting airways. Subsequently, SOX9^+ tip progenitors produce progeny for forming saccules. Production of the conducting airways and saccules is also closely associated with proper cell differentiation. Club cells (CC10^+), ciliated cells (Ac-tub^+) and goblet cells line the conducting airways while saccules contain AT1 (HOPX^+; T1α^+) and AT2 cells (SPC^+) cells. SOX9^+ progenitors differentiate into HOPX^+ cells that are adjacent to the SOX2^+ conducting airways. SOX9^+ progenitors are located distal to HOPX^+ cells and differentiate into SPC^+ cells within saccules. Disruption of mTORC1 signaling leads to reduced mitochondrial capacity. SOX9^+ tip progenitors expand to generate SOX9^+ transition progenitors. However, transition progenitors fail to produce SOX2^+ progeny and thus the conducting airways are missing. Instead, lung cysts are generated. SOX9^+ progenitors produce saccule-like structures on the surface or the edge of lung cysts. All values are mean ± SEM. (*) *p* < 0.05; (**) *p* < 0.01; (***) *p* < 0.001; ns not significant (two-tailed, unpaired Student's *t*-test). Source data are provided as a Source Data file.

GTT-3′, *Hk2* forward, 5′-GCCAGCCTCTCCTGATTTTAGTGT-3′; reverse, 5′-GGGAACACAAAAGACCTCTTCTGG-3′.

## Western blotting analysis

Purified lung epithelial cells were lysed in 2x Western blotting buffer (120 mM Tris-Cl pH 6.8, 4% SDS, 20% Glycerol, 200 mM DTT, 0.02% Bromophenol Blue). The lysates were centrifuged at 16,100 x *g* for 15 min at 4 °C before loading onto SDS-PAGE. The primary antibodies used were rabbit anti-Raptor (1:2000, EMD Millipore Corporation, Cat# 09-217; RRID:AB_1659713), mouse anti-S6 ribosomal protein (1:2000, Cell Signaling Technology, Cat# 2317; RRID:AB_2238583), rabbit anti-phospho-S6 ribosomal protein (Ser235/236) (1:2000, Cell Signaling Technology, Cat# 4856), rabbit anti-MLC2 (1:500; Cell Signaling Technology, Cat#3672; RRID:AB_10692513), goat anti-TFAM (1: 250, Santa Cruz Biotechnology, Cat# sc-23588; RRID:AB_2303230), rabbit anti-COX10 (1:500, Proteintech, Cat# 10611-2-AP; RRID:AB_2084833), mouse anti-MTCO1 (1:500, Abcam, Cat# ab14705; RRID:AB_2084810), and mouse anti-alpha-tubulin (1:3000, Developmental Studies Hybridoma Bank, Cat# 12G10; RRID:AB_1157911). Western blotting images were captured on a LI-COR Odyssey Fc device. The uncropped images are available in the Source Data file.

## RNA-Seq analysis

Bulk RNA-Seq of mouse lungs was performed as previously described[31]. In brief, embryonic lungs from *Rptor^f/f; Shh^Cre/+*, *Tfam^f/f; Shh^Cre/+* and *Cox10^f/f; Shh^Cre/+* mice and their littermate controls (*Rptor^f/+; Shh^Cre/+*, *Tfam^f/+; Shh^Cre/+* and *Cox10^f/+; Shh^Cre/+* mice) were collected at 14.5 *dpc*. Lungs from *Rptor^f/f; Shh^Cre/+* mice and controls were also collected at 11.5 *dpc*. For lungs collected at 11.5 *dpc*, three or four lungs of the same genotype were combined to reach a sufficient amount of RNA for RNA-Seq. The samples were lysed in 500 μl TRIzol (Ambion, Cat# 15596018). After adding 100 μl chloroform, the samples were centrifuged at 4 °C for 15 min. The upper aqueous phase was mixed with an equal volume of 70% ethanol. RNA was extracted with the RNeasy Mini Kit (Qiagen, Cat# 74104) following the manufacturer's instructions. After quality evaluation with an Agilent 2100 Bioanalyzer, the RNAs were then sequenced on BGISEQ-500 sequencer. Differential gene expression, gene ontology (GO) enrichment analyses and the barplot of gene ontology enrichment were performed with RStudio (R version 3.4.0). Heatmap images were generated using the online Heatmapper software [http://www.heatmapper.ca/]. Datasets have been deposited into NCBI's Gene Expression Omnibus database and

are accessible through GEO Series accession numbers GSE189327 and GSE213202.

## Statistics and reproducibility

All the quantification including the cell number, diameter, thickness, distance and area of the lungs and the signal density of immunostaining were performed using ImageJ (Version 1.49). Measurement of the relative diameter of the branches and the relative thickness of the mesenchyme relied on the GFP and tdTomato signals, respectively, produced in *Shh^Cre/+; ROSA26^mTmG/+* and *Rptor^f/f; Shh^Cre/+; ROSA26^mTmG/+* lungs. For each lung, at least 5 distal buds were used for assessing the diameter and thickness. For detecting the signal density of immunostaining, at least 15 views of images (5 views per biological repeat) were measured. The images were taken under 40X magnification using a Nikon Eclipse E1000 Microscope with the same exposure.

The mutant phenotypes described in this study were completely penetrant. All the mice of the same genotype showed similar, if not identical, lung phenotypes at each timepoint and there was little variation between the mutant lungs. The mutant phenotypes were apparent and could be discerned by the naked eye or under the dissecting scope. Accordingly, changes in morphology and molecular markers in the mutants can also be readily distinguished from the controls under the microscope. In this regard, blinding and randomization are not applicable. Nevertheless, sample tissues or sections were randomly selected for analysis such as mtDNA quantification, EdU counting, ATP production and RNA-Seq.

For all the micrograph-related assays, at least three biological repeats were performed. The completely penetrant and invariant phenotypes have allowed us to provide representative images shown in the figures. It has also allowed us to obtain statistical significance with the number of mice used in this study. Since an average of two mutants were obtained in each litter, we combined mutants from different litters for statistical analysis. Again, the completely penetrant and invariant phenotypes facilitated our statistical analysis.

For all the biological repeats, greater than or equal to three repeats were performed. The n number for biological replicates are shown in the figure legends. All the graphs were generated using GraphPad Prism 7 and the statistics are shown as mean value ± SEM. Two-tailed Student's *t*-tests were performed to assess the P values of the comparisons between the control and mutant groups; the statistical significance was evaluated as *$P < 0.05$; **$P < 0.01$; ***$P < 0.001$.

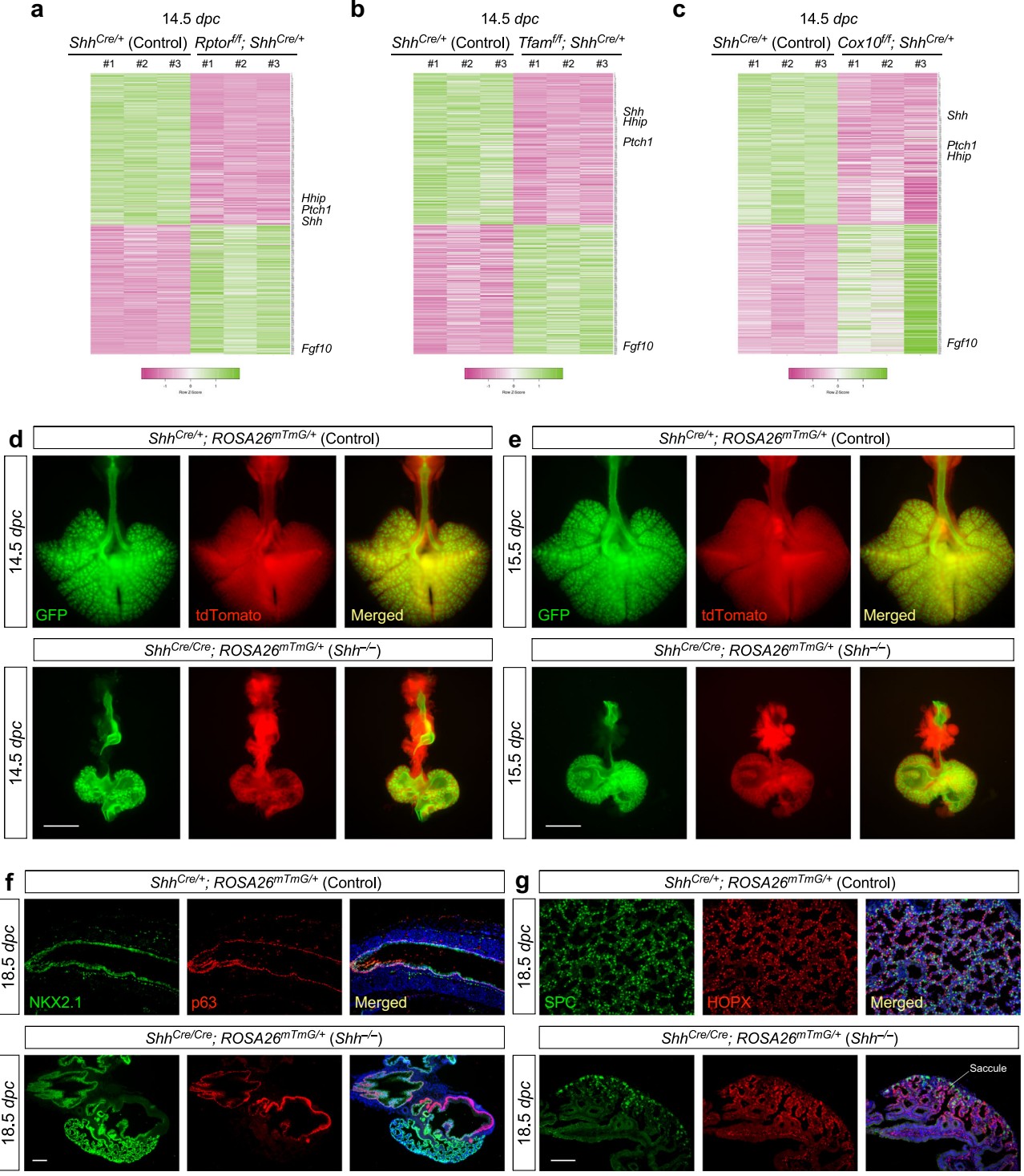

**Fig. 9 | Defective Hh signaling perturbs the development of the conducting airways without affecting saccule formation. a**, **b**, **c** Heatmap of gene expression from control (*Shh^Cre/+*), *Rptor^f/f*; *Shh^Cre/+*, *Tfam^f/f*; *Shh^Cre/+* and *Cox10^f/f*; *Shh^Cre/+* lungs at 14.5 *days post coitus* (*dpc*). 641 genes that showed similar differential gene expression in all three datasets were plotted. The relative positions of Hh pathway components, *Shh*, *Ptch1* and *Hhip*, and *Fgf10* on the gene list were indicated on the heatmap. **d**, **e** Whole-lung imaging of dissected lungs from *Shh^Cre/+*; *ROSA26^mTmG/+* (control) and *Shh^Cre/Cre*; *ROSA26^mTmG/+* (*Shh^−/−*) mice at 14.5 and 15.5 *dpc*. GFP was activated from the *ROSA26^mTmG* allele in all lung epithelial cells by *Shh^Cre*; tdTomato marked all non-epithelial cells. Many GFP⁺ buds were present on the cystic surface of *Shh^Cre/Cre*; *ROSA26^mTmG/+* (*Shh^−/−*) lungs. Scale bar = 1 mm. **f** Immunostaining of lung

sections collected from *Shh^Cre/+*; *ROSA26^mTmG/+* (control) and *Shh^Cre/Cre*; *ROSA26^mTmG/+* (*Shh^−/−*) lungs at 18.5 *dpc*. NKX2.1 labeled all epithelial cells, while p63 marked the basal cells in trachea and main stem bronchi. The cystic wall in *Shh^Cre/Cre*; *ROSA26^mTmG/+* (*Shh^−/−*) lungs contained cells that expressed markers of trachea/main stem bronchi. Scale bar = 100 μm. **g** Immunostaining of lung sections collected from *Shh^Cre/+*; *ROSA26^mTmG/+* (control) and *Shh^Cre/Cre*; *ROSA26^mTmG/+* (*Shh^−/−*) lungs at 18.5 *dpc*. SPC labeled alveolar epithelial type II (AT2) cells, while HOPX marked alveolar epithelial type I (AT1) cells. Saccule-like structures were found on the cystic surface of *Shh^Cre/Cre*; *ROSA26^mTmG/+* (*Shh^−/−*) lungs and contained AT1 and AT2 cells. Scale bar = 100 μm.

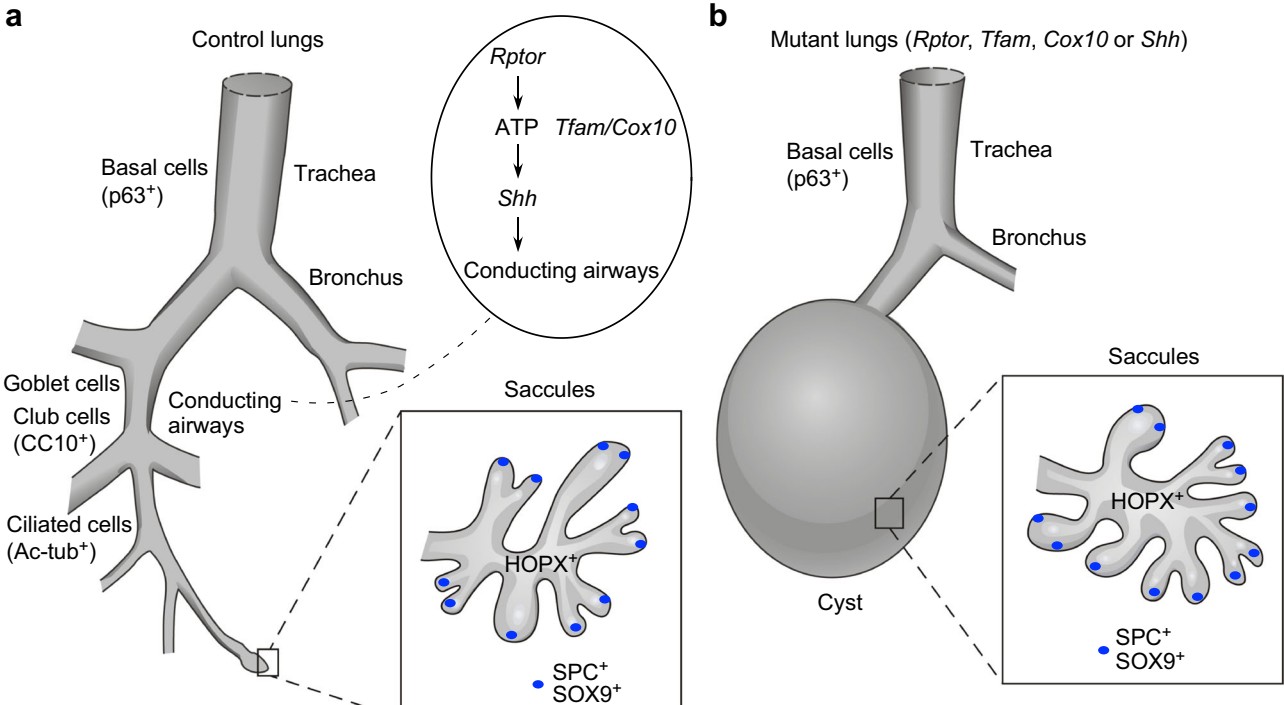

**Fig. 10 | A model of lung branching and saccule formation. a** Schematic diagram of lung branching and saccule formation in control lungs. The conducting airways extend from the trachea and main stem bronchi. A signaling cascade that involves mTORC1 (*Rptor*) signaling, mitochondrial capacity (*Tfam* and *Cox10*) and Hedgehog (Hh) signaling controls lung branching to form the conducting airways. Saccules are subsequently generated at the end of the conducting airways. Production of the conducting airways and saccules is also closely associated with proper cell differentiation. **b** Schematic diagram of lung branching and saccule formation in lungs with disrupted mTORC1 signaling, reduced mitochondrial capacity or perturbed Hedgehog (Hh) signaling. In these mutants, the conducting airways are missing and are replaced by lung cysts. Saccule-like structures form on the cystic surface. This suggests that distinct programs control lung branching and saccule formation. It also indicates the presence of a developmental clock that controls lung development.

## Reporting summary

Further information on research design is available in the Nature Portfolio Reporting Summary linked to this article.

## Data availability

All the related data for this study are available in the published article and the Supplementary Information file. Additional data that support the findings of this study are available from the corresponding authors upon reasonable request. Raw and analyzed data of RNA-Seq have been deposited to the Gene Expression Omnibus (GEO) database under accession numbers GSE189327 and GSE213202. Source data are provided with this paper.

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

## Acknowledgements

Some data for this study were acquired at the Nikon Imaging Center at CVRI. This work was supported by grants (R01 HL142876) from the National Institutes of Health to P.-T.C.

## Author contributions

K.Z., E.Y. and P.-T.C. conceived and designed the study. K.Z., E.Y., E.C., B.C. and P.-T.C. performed the experiments. K.Z., E.Y., E.C., B.C., E.Y.C. and P.-T.C analyzed the data. K.Z., E.Y., and P.-T.C. wrote the manuscript. P.-T.C. supervised the research.

## Competing interests

The authors declare no competing interests.
