## [Peer Review File · Nature Communications]

mTORC1 signaling facilitates differential stem cell differentiation to shape the developing murine lung and is associated with mitochondrial capacityREVIEWER COMMENTS

Reviewer #1 (Remarks to the Author):

This is a comprehensive study looking at the potential role of proteins regulating mitochondria capacity in lung development. A series of mouse mutants were analyzed coupled with characterization through immunostaining and RNA sequencing. The authors conclude that these proteins are important for modulating lung branching morphogenesis. Their findings further suggest decoupling of branching morphogenesis and sacculle formation. These results offer genetic evidence that associates mitochondria capacities and lung branching. With that being said, this study can be further enhanced with mechanistic investigations. At current form, the authors mostly focused on the phenotypic characterization.

Several potentially interesting points are covered in this study, for example, proximal-distal transition and AT2 cell differentiation associated with Sox9+ progenitor cells. Is there a possible connection between mitochondria capacity and these biological events at the molecular level? In addition, will these proteins (Raptor, Lrprrc, Tfam, Cox10) have common downstream target(s) that modulate branching morphogenesis? How is energy production related to branching morphogenesis at molecular level? These could be some of mechanistic questions that the authors may consider.

The approach used to assess mitochondria capacity can be improved, for example, the authors used the whole lung (at E13.5) to measure the relative ratio of mitochondrial DNA (mtDNA) to nuclear DNA (nDNA) (Fig5D, 6Q,R). Can the authors purify epithelium for this assessment?

Additionally, more rigorous quantification should be considered when comparing mutants and controls. For example, in Fig6S, the numbers of mitochondria and golgi should be quantified.

Minor issue:

1. MPC1 staining seems unspecific in the figure S6, especially for S6H. It is enriched in the apical domain of almost all airway epithelial cells.
2. This reviewer thinks that the authors may want to discuss their findings in reference to others' findings e.g. Proc Natl Acad Sci U S A. 2013 Nov 5;110(45):18042-51. and Proc Natl Acad Sci U S A. 2013 Nov 19;110(47):E4456-64.

Reviewer #2 (Remarks to the Author):

What are the noteworthy results?

This manuscript employs several murine floxed gene models to report a novel role for mTORC1 in the regulation of epithelial differentiation during lung development. Key observations are that,

- Rptor knockout in airway epithelium abolishes RPS6 phosphorylation, a widely used reporter of mTORC1 activity
- This was associated with perturbed epithelial branching and thickening of non-epithelial cells which occurred with no change in epithelial proliferation
- Decreased epithelial polarization and deposition of parabronchial smooth muscle.

These changes in lung morphology were linked to disrupted transition of SOX9-SOX2 proximal-distal differentiation, but not SOX9+ cell expansion and sacculle formation since, despite changes in cell specification along the proximal airway, sacculle formation and alveolar type I and II cell differentiation continued in the distal lung sacculles.

Heatmap analysis revealed deficiencies in glycolytic and related pathways in the presence of Raptor knockout with significant metabolic perturbation confirmed by a reduction in mtDNA copy number and

ATP production in Rptor-deficient lungs. Floxed knockout of Tfam or Cox10 captured the general phenotype produced by Rptor knockout, with some differences.

The authors conclude that selective use of energy by different developmental processes drives the branching morphogenesis program and that an axis of signaling which links mTORC1-metabolic pathways-Sox2-Sox9 differentiation is chiefly responsible for this process.

Will the work be of significance to the field and related fields? How does it compare to the established literature? If the work is not original, please provide relevant references.

Interest in the role of mTORC1 has developed considerably since earlier in vitro studies established evidence that activation of this pathway by over-expression of rheb altered patterns of explant growth and fetal distal lung epithelial cell responsiveness to FGF-10. The major contribution of the current manuscript is to take this research in vivo for the first time using floxed knockdown of Rptor and other genes. This novel approach represents a valuable step forward for deciphering mTOR complex function in the branching morphogenesis program.

Does the work support the conclusions and claims, or is additional evidence needed?

Additional evidence is needed - please see comments below. In addition, Nature Communications require that the key aims and conclusions of their published articles are compelling, logical and clear. The purpose of the current manuscript seems scattered between 1) role of mTORC1 in branching morphogenesis, 2) energy dependence of cell processes in branching routines and 3) the value of the technical approach. Specifically, do the authors intend the primary focus of the manuscript to be the report of a new metabolic paradigm of pulmonary morphogenesis or are they showcasing an analytical approach for investigating morphogenic/metabolic gene function in lung? For example, in p4 parag 1 the authors state: *"In this study, we have employed lung development in mice as a system to decipher energy dependence in different cell types and cellular processes in vivo."* but the concluding model given in Figure 9 does not appear to metabolic detail stratified by cell behaviors along the airway. The concluding sentences on this same paragraph and at the end of the discussion (p14) both highlight the value of this technical approach as a framework for future investigation rather than emphasizing the molecular/metabolic processes which have been identified. Do the authors intend their technical approach to be the main value of the manuscript?

Is the methodology sound? Does the work meet the expected standards in your field?

Demonstration of Objectivity: Many of the results are presented as single images selected as representative of n=3-5 without explanation as to why this is considered representative of the experimental population. Although this has become accepted practice in the field, there is need to address wider perceptions about science reproducibility by demonstrating that studies have been objectively designed and are resilient to challenge. Dongen and Sikorski address this issue in "Objectivity for the Research Worker" European Journal for Philosophy of Science (2021) 11: 93; (doi.org/10.1007/s13194-021-00400-6) in which they conclude: *"...scientific practice becomes more objective when it becomes demonstrably more resilient to actions and decisions that have the potential to influence its outcome; concretely, when:*

- a) the study design and data collection becomes demonstrably more resilient to the scientists' influence on the data;
- b) and the data processing and analysis become demonstrably more resilient to ad hoc decision making and selective reporting of positive results".

This manuscript does not satisfy these criteria in its current form because there is no mention of steps taken to minimize bias in the conduct of the experiments or analysis of the data. For example, there is no use of randomization, blinding independent statistical analysis or determination of study power, data distribution or quantification of image-based data. "n" is not defined in the manuscript (eg were the experiments conducted using independent litters? In statistical terms, why was n=3 considered an

appropriate lower value? None of the studies report n greater than 5 – why was this considered an appropriate upper limit?). The concluding sentence of the manuscript states that *“Our work provides a framework for further studies.....by combining live imaging, organ culture and molecular tools”*. The authors are encouraged to strengthen the value of their proposed framework by adding procedures which show the study to be demonstrably objective in its use of these tools. As a Nature Communications article, this should set the standard for the rest of the field.

Are there any flaws in the data analysis, interpretation and conclusions? Do these prohibit publication or require revision?

1. Validation of the loss of raptor signal and other knockdown genotypes: Decline in raptor expression is inferred from loss of the epithelial IF signal for mTORC1-dependent phosphorylation of RPS6 but there is no alternative validation of the *Rptor* null genotype or any other demonstration that mTORC1 signaling is disrupted. At this time there is consequently no validation that the floxed *Rptor* approach has nulled raptor mRNA and protein expression beyond absence of the IF RPS6 signal. This information is important to report penetrance of the knockdown genotype and to control, for example, Cre mosaicism, and differences in efficiency of recombination from one gene to another.

2. The coincident effects of mTORC1 inhibition and metabolic disruption is an interesting observation but does not necessarily imply that the developmental outcomes arise purely from metabolic disturbance. Given the close links between mTORC1, autophagy and mitochondrial function, it is very surprising that no attention has been paid to the alternative hypothesis that dysregulation of autophagy drives the perturbation in SOX9-SOX2. The manuscript is incomplete without discussion of this signaling axis.

3. Control for mTORC2 effects: mTORC1 is a repressor of mTORC2 signaling and so its inhibition promotes compensatory mTORC2 effects in some systems. The possibility that anomalous mTORC2 activation could influence morphological patterning is not addressed experimentally or discussed in the manuscript.

Is there enough detail provided in the methods for the work to be reproduced?

Yes

Reviewer #3 (Remarks to the Author):

Zhang et al NC review March 2022 DR

Zhang et al. knocked out Raptor in the mouse lung epithelium using ShhCre. Found defects in branching morphogenesis, but no changes in the trachea or main stem bronchus. No new daughter branches formed, and the mice died after birth with respiratory failure. Sox2+ cells were absent from the 5 main branches that arise from the primary buds at ~E11.5 and remained confined to the trachea and main stem bronchi as the Raptor KO mice age. Sox2+ cells normally differentiate from Sox9+ cells, with Sox9 giving rise to AT1/2 cells and Sox2 giving rise to goblet/club/ciliated cells in the proximal airway. Therefore, these results suggest that Sox9 progenitors require mTORC1 to produce Sox2+ cells, resulting in loss of conducting airways in Raptor-KO lungs. To understand the mechanisms through which mTORC1 controls lung branching, the authors studied mitochondria and found a decrease in whole lung mtDNA/nDNA ratio of about 30% (Fig 5D,E) and ~25% less ATP production. To test whether this is the mechanism in vivo, they knocked out Tfam and Cox10 using ShhCre. The MtDNA/nDNA ratio was decreased by ~50% in the Tfam KO mice, as expected, and ATP production was decreased by ~40% in both models. Defects in branching morphogenesis were found in these mice, similar to the Raptor KO, and similar defects in the distribution of Sox2+ cells. Saccule

formation by the Sox9+ cells appeared to be intact.

It is clear from the data that mTORC1 plays a crucial role in branch formation within the developing lung, and that that mitochondrial DNA and ATP production are decreased in the raptor knockout lung, but there is not strong evidence to link these two observations. While the mice with impaired mitochondria develop a similar lung phenotype to the raptor KO mice, this similarity does not imply causation. In addition to this concern, many of the data rely on IF images. Quantitative metrics of the imaging data are needed along with protein and mRNA analyses to validate the microscopic observations.

Major comments:

- 1) The term "tubular organ" in the title is an unusual choice – replace with developing murine lung? Also in the title, it cannot be concluded based on the data that mTORC1 is acting via mitochondrial capacity – this an interesting hypothesis but not proven by the available data.
- 2) While the IF data are beautiful and informative, the lack of western blot and qPCR analyses is surprising. The authors could take advantage of the mTmG system to sort their tissues by FACS. They could have used FACS to isolate GFP+, CRE+ cells prior to western blotting, qPCR, or even RNAseq.
- 3) A more quantitative approach to analyze the lung development phenotypes would strengthen the conclusions considerably. Currently, the data are largely qualitative. How many individual mice were dissected and immuno-stained? Did all the mice show similar development at each timepoint and what was the range of variation between mice of the same genotype? What measures were taken to ensure that images were taken from the same exact region in control and raptor fl/fl lungs? Were any measurements taken of lung mass or volume? Perhaps the authors could quantitate the length and number of lung branches at each time point.
- 4) 1A-F: Is pRPS6 decreased in the raptor fl/fl mice, or are total levels decreased? Additional IF staining for total RPS6, and/or a western blot, are needed. The same is true for pMLC2 in figure S1.
- 5) 1X-K: Because the structure of these tissues varies so much between control and raptor fl/fl mice, IF alone cannot be used to evaluate expression of F-Actin and smooth muscle actin. If western blotting or qPCR are not possible, the authors might be able to normalize the fluorescent signal to total area of the tissue. This can be done in ImageJ.
- 6) 1S-W: How was the Edu stain quantified? Was the Edu signal normalized to DAPI signal?
- 7) 2O-V: The introduction states that cessation of Sox9 expression is concomitant with Sox2 activation, but there is significant overlap of SOX2 and SOX9 expression in these images.
- 8) 2W-H': Quantification is necessary to conclude that the Sox9 -> Sox2 transition is impaired in raptor fl/fl mice. Perhaps a ratio of RFP signal to GFP signal, normalized to total area of the lungs, could be used.
- 9) 3A-H: please include the RFP channel alone
- 10) 4: Please provide quantification of the fluorescent signal relative to DAPI.
- 11) 5A,B,C: The authors chose to focus on the difference in mitochondrial content activity between control and raptor fl/fl mice, which is surprising given that no mitochondrial pathways came up in the GO analyses. This should be discussed.
- 12) 5, 6, 8: It is unclear whether there is less MTOC expressed in each of the knockout models, relative to control mice. There is a decrease in GFP signal, but this decrease could be due to the increase in cystic "empty" space. For a proper comparison, quantification to GFP signal relative to tissue area, in multiple sections, should be performed.

Minor comments:

- 1) Legend for Fig 6 A-P includes the presumed mechanisms through which MTCO1 is reduced in TFAM and Cox10 KO (reduced transcription of MTCO1; reduced MTCO1 protein stability, etc). Since the authors did not generate these data, this information should perhaps be in the manuscript itself and not the figure legend.
- 2) Legend for Fig 9A – progenies should be progeny.
- 3) Model, Figure 9A – is there evidence that mTORC1/mitochondria contribute to the differentiation of

Sox9+ cells into AT1/AT2? There is a "thin" arrow here.

4) Legend for 9B – what is meant by "in contrast" – in contrast to what?

5) For the quantification of EdU positive cells (Fig 1W and elsewhere) how was the number of EdU positive cells measured? This does not seem to be in the legend or methods.

6) Figure S6: typo in title (reduction misspelled)

Reviewer #1 (Remarks to the Author):

This is a comprehensive study looking at the potential role of proteins regulating mitochondria capacity in lung development. A series of mouse mutants were analyzed coupled with characterization through immunostaining and RNA sequencing. The authors conclude that these proteins are important for modulating lung branching morphogenesis. Their findings further suggest decoupling of branching morphogenesis and saccule formation. These results offer genetic evidence that associates mitochondria capacities and lung branching. With that being said, this study can be further enhanced with mechanistic investigations. At current form, the authors mostly focused on the phenotypic characterization.

Several potentially interesting points are covered in this study, for example, proximal-distal transition and AT2 cell differentiation associated with Sox9+ progenitor cells. Is there a possible connection between mitochondria capacity and these biological events at the molecular level? In addition, will these proteins (Raptor, Lrp1rc, Tfam, Cox10) have common downstream target(s) that modulate branching morphogenesis? How is energy production related to branching morphogenesis at molecular level? These could be some of mechanistic questions that the authors may consider.

The reviewer raised an important question. We have performed RNA-Seq analysis of *Raptor*^{ff}; *Shh*^{Cre/+}, *Tfam*^{ff}; *Shh*^{Cre/+} and *Cox10*^{ff}; *Shh*^{Cre/+} lungs at 14.5 days post coitus (dpc) to try to identify common downstream targets and establish their connections (new Fig. 9). We found that the Hedgehog (Hh) signaling pathway was downregulated in the lungs of all three mutants (new Fig. 9). Moreover, KEGG pathway analysis revealed that several pathways were perturbed in the mutant lungs (new Supplementary Fig. 15), including the Rap1 signaling pathway, the PI3K-Akt signaling pathway and the Ras signaling pathway.

We further showed that the conducting airways failed to develop in *Shh* mutant lungs while saccule formation still proceeded (new Fig. 9). This is similar to what we observed in *Raptor*-, *Tfam*- and *Cox10*-deficient lungs in this manuscript. We also noted that Kras activation in the lung epithelium was reported to disrupt the SOX9 to SOX2 transition (Chang et al. *Proc Natl Acad Sci*, 2013), a common defect shared by *Raptor*-, *Tfam*- and *Cox10*-deficient lungs. Taken together, we propose that mTORC1 signaling functions in a signaling network, part of which involves mitochondrial capacity, and Hh and Kras signaling. These observations provide new molecular insight into how mTORC1 signaling, mitochondrial capacity and the downstream signaling events control branching morphogenesis. Nevertheless, the detailed molecular mechanisms by which mitochondrial capacity and ATP production regulate branching require additional investigations.

The approach used to assess mitochondria capacity can be improved, for example, the authors used the whole lung (at E13.5) to measure the relative ratio of mitochondrial DNA (mtDNA) to nuclear DNA (nDNA) (Fig5D, 6Q,R). Can the authors purify epithelium for this assessment?

As suggested by the reviewer, we have purified lung epithelial cells using anti-EpCAM antibody from control and *Raptor*, *Tfam* and *Cox10* mutant mouse lungs and conducted quantitative analyses. They include the relative ratio of mitochondrial DNA to nuclear DNA (new Fig. 5d, e and Fig. 6e, f), relative ATP production (new Fig. 5f and Fig. 6g) and Western blotting analysis of pathway components (new Supplementary Fig. 1c and Supplementary Fig. 10a).

Additionally, more rigorous quantification should be considered when comparing mutants and controls. For example, in Fig6S, the numbers of mitochondria and golgi should be quantified.

We have conducted more rigorous quantifications on control and mutant lungs. They are shown in new Fig. 1b, d, g, h, Fig. 2b, Fig. 4e-i, Fig. 5h, j, Fig. 6c-d, Fig. 8c-e, Supplementary Fig. 1a, b, Supplementary Fig. 3b, d, Supplementary Fig. 6e, Supplementary Fig. 7b, d, f, Supplementary Fig. 8b, d, and Supplementary Fig. 9c.

Mitochondria and the Golgi apparatus are dynamic organelles. It is challenging to quantify the exact number of mitochondria and the Golgi apparatus. Nevertheless, we have utilized the established methods to measure mtDNA/nDNA ratio (new Fig. 5d, e) and the signals of MPC1 and MTCO1 (new Fig. 5h, j) to investigate changes in mitochondrial content. We have also quantified GM130 (a cis-Golgi matrix protein and a marker of the Golgi apparatus) in control and *Rptor* mutant lungs (new Supplementary Fig. 9c).

Minor issue:

1. MPC1 staining seems unspecific in the figure S6, especially for S6H. It is enriched in the apical domain of almost all airway epithelial cells.

We have provided higher resolution images of MPC1 (new Supplementary Fig. 9a), in which MPC1 expression was relatively ubiquitous in the cytosol with more signals at the apical surface of epithelial cells. Perhaps due to the optical section and resolution, the original images were somewhat misleading and thus have been replaced.

2. This reviewer thinks that the authors may want to discuss their findings in reference to others' findings e.g. Proc Natl Acad Sci U S A. 2013 Nov 5;110(45):18042-51. and Proc Natl Acad Sci U S A. 2013 Nov 19;110(47):E4456-64.

As suggested by the reviewer, we have added more discussions on the two *PNAS* papers in the context of our findings.

Reviewer #2 (Remarks to the Author):

What are the noteworthy results?

This manuscript employs several murine floxed gene models to report a novel role for mTORC1 in the regulation of epithelial differentiation during lung development. Key observations are that,

- Rptor knockout in airway epithelium abolishes RPS6 phosphorylation, a widely used reporter of mTORC1 activity*
- This was associated with perturbed epithelial branching and thickening of non-epithelial cells which occurred with no change in epithelial proliferation*
- Decreased epithelial polarization and deposition of parabronchial smooth muscle.*

These changes in lung morphology were linked to disrupted transition of SOX9-SOX2 proximal-distal differentiation, but not SOX9+ cell expansion and saccule formation since, despite changes in cell specification along the proximal airway, saccule formation and alveolar type I and II cell differentiation continued in the distal lung saccules.

Heatmap analysis revealed deficiencies in glycolytic and related pathways in the presence of Raptor knockout with significant metabolic perturbation confirmed by a reduction in mtDNA copy number and ATP production in Rptor-deficient lungs. Floxed knockout of Tfam or Cox10 captured the general phenotype produced by Rptor knockout, with some differences.

The authors conclude that selective use of energy by different developmental processes drives the branching morphogenesis program and that an axis of signaling which links mTORC1-metabolic pathways-Sox2-Sox9 differentiation is chiefly responsible for this process.

Will the work be of significance to the field and related fields? How does it compare to the established literature? If the work is not original, please provide relevant references.

Interest in the role of mTORC1 has developed considerably since earlier in vitro studies established evidence that activation of this pathway by over-expression of rheb altered patterns of explant growth and fetal distal lung epithelial cell responsiveness to FGF-10. The major contribution of the current manuscript is to take this research in vivo for the first time using floxed knockdown of Rptor and other genes. This novel approach represents a valuable step forward for deciphering mTOR complex function in the branching morphogenesis program.

Does the work support the conclusions and claims, or is additional evidence needed?

Additional evidence is needed - please see comments below. In addition, Nature Communications require that the key aims and conclusions of their published articles are compelling, logical and clear. The purpose of the current manuscript seems scattered between 1) role of mTORC1 in branching morphogenesis, 2) energy dependence of cell processes in branching routines and 3) the value of the technical approach. Specifically, do the authors intend the primary focus of the manuscript to be the report of a new metabolic paradigm of pulmonary morphogenesis or are they showcasing an analytical approach for investigating morphogenic/metabolic gene function in lung? For example, in p4 parag 1 the authors state: "In this study, we have employed lung development in mice as a system to decipher energy dependence in different cell types and cellular processes in vivo." but the concluding model given in Figure 9 does not appear to metabolic detail stratified by cell behaviors along the airway. The concluding sentences on this same paragraph and at the end of the discussion (p14) both highlight the value of this technical approach as a framework for future investigation rather than emphasizing the molecular/metabolic processes which have been identified. Do the authors intend their technical approach to be the main value of the manuscript?

The main purpose of the manuscript is to understand the molecular basis of pulmonary morphogenesis from the perspective of mTORC1 signaling and mitochondrial function/ATP production. The technical approach serves as a tool to reveal the molecular basis of pulmonary morphogenesis. In response to the reviewer's comments, we have modified the text to highlight the main progress reported in the manuscript.

In this study, we established that disruption of *Rptor* (mTORC1) led to lung branching defects in mice. Through molecular analysis of *Rptor* mutant lungs, we showed that mitochondria capacity/ATP production function as downstream effectors of mTORC1 during lung branching. This notion was supported by genetic studies in which disruption of mitochondrial function (e.g.,

Tfam or *Cox10* mutants) recapitulates lung defects in *Rptor* mutants. Our transcriptome analysis of the mutant lungs identified Hedgehog (Hh) signaling as a downstream event of mTORC1/mitochondria capacity/ATP production. This model was substantiated by branching defects in *Shh* mutant lungs, similar to those in *Rptor*, *Tfam* and *Cox10* mutant lungs. Overall, the logic that reaches our conclusions is clear and the data are compelling.

Moreover, our genetic and molecular analysis revealed the unexpected finding that sacculle formation proceeded at the proper developmental time in the absence of lung branching. This suggests that distinct programs control lung branching and sacculle formation. It also indicates the presence of a developmental clock. Our results establish the foundation for identifying the program that regulates sacculle formation and the developmental clock that controls lung development.

These key points are summarized in models (new Fig. 8f and Fig. 10).

Is the methodology sound? Does the work meet the expected standards in your field?

Demonstration of Objectivity: Many of the results are presented as single images selected as representative of n=3-5 without explanation as to why this is considered representative of the experimental population. Although this has become accepted practice in the field, there is need to address wider perceptions about science reproducibility by demonstrating that studies have been objectively designed and are resilient to challenge. Dongen and Sikorski address this issue in “Objectivity for the Research Worker” European Journal for Philosophy of Science (2021) 11: 93; (doi.org/10.1007/s13194-021-00400-6) in which they conclude: “...scientific practice becomes more objective when it becomes demonstrably more resilient to actions and decisions that have the potential to influence its outcome; concretely, when:

a) the study design and data collection becomes demonstrably more resilient to the scientists’ influence on the data;

b) and the data processing and analysis become demonstrably more resilient to ad hoc decision making and selective reporting of positive results”.

This manuscript does not satisfy these criteria in its current form because there is no mention of steps taken to minimize bias in the conduct of the experiments or analysis of the data. For example, there is no use of randomization, blinding independent statistical analysis or determination of study power, data distribution or quantification of image-based data. “n” is not defined in the manuscript (eg were the experiments conducted using independent litters? In statistical terms, why was n=3 considered an appropriate lower value? None of the studies report n greater than 5 – why was this considered an appropriate upper limit?). The concluding sentence of the manuscript states that “Our work provides a framework for further studies.....by combining live imaging, organ culture and molecular tools”. The authors are encouraged to strengthen the value of their proposed framework by adding procedures which show the study to be demonstrably objective in its use of these tools. As a Nature Communications article, this should set the standard for the rest of the field.

The mutant phenotypes described in this study were completely penetrant. All the mice of the same genotype showed similar, if not identical, lung phenotypes at each timepoint and there was little variation between the mutant lungs. The mutant phenotypes were apparent and could be discerned by the naked eye or under the dissecting scope. Accordingly, changes in morphology and molecular markers in the mutants can also be readily distinguished from the controls under

the microscope. In this regard, blinding and randomization are not applicable. Nevertheless, sample tissues or sections were randomly selected for analysis such as mtDNA quantification, EdU counting, ATP production and RNA-Seq.

The completely penetrant and invariant phenotypes have allowed us to provide representative images shown in the figures. It has also allowed us to obtain statistical significance with the number of mice used in this study. Of note, we have analyzed a far larger number of control and mutant lungs over a period of several years. The n number shown in the manuscript simply represents a fraction of the animals on which a thorough analysis was conducted. Since an average of 2 mutants were uncovered in each litter, we combined mutants from different litters for statistical analysis. Again, the completely penetrant and invariant phenotypes facilitated our statistical analysis. We have clarified these points in the revised manuscript.

Are there any flaws in the data analysis, interpretation and conclusions? Do these prohibit publication or require revision?

1. Validation of the loss of raptor signal and other knockdown genotypes: Decline in raptor expression is inferred from loss of the epithelial IF signal for mTORC1-dependent phosphorylation of RPS6 but there is no alternative validation of the Rptor null genotype or any other demonstration that mTORC1 signaling is disrupted. At this time there is consequently no validation that the floxed Rptor approach has nulled raptor mRNA and protein expression beyond absence of the IF RPS6 signal. This information is important to report penetrance of the knockdown genotype and to control, for example, Cre mosaicism, and differences in efficiency of recombination from one gene to another.

We have purified lung epithelial cells using anti-EpCAM antibody from control and *Rptor*, *Tfam* and *Cox10* mutant mouse lungs. Western blotting analysis of isolated epithelial cells confirmed the loss of RPTOR, TFAM and COX10 in the mutants (new Supplementary Figure 1c and Supplementary Figure 10a). These results show that the Cre line used in this study is efficient; thus the resulting phenotypes are completely penetrant as mentioned above.

2. The coincident effects of mTORC1 inhibition and metabolic disruption is an interesting observation but does not necessarily imply that the developmental outcomes arise purely from metabolic disturbance. Given the close links between mTORC1, autophagy and mitochondrial function, it is very surprising that no attention has been paid to the alternative hypothesis that dysregulation of autophagy drives the perturbation in SOX9-SOX2. The manuscript is incomplete without discussion of this signaling axis.

We agree with the reviewer that mTORC1 has multiple functions. It is interesting to note that a connection established in cell lines is not necessarily recapitulated in the animals. For instance, perturbation of autophagy has no apparent effects on lung branching as shown in previous publications (e.g., Cheong et al. *Autophagy*, 2014; Pyo et al. *Nat Commun*, 2013). Therefore, we have focused on downstream events linked to mTORC1 that also play an important role in lung branching. This point has been clarified in the revision.

3. Control for mTORC2 effects: mTORC1 is a repressor of mTORC2 signaling and so its inhibition promotes compensatory mTORC2 effects in some systems. The possibility that

anomalous mTORC2 activation could influence morphological patterning is not addressed experimentally or discussed in the manuscript.

We have removed *Rictor* (mTORC2) in the lung epithelium using *Sox9-Cre* and did not observe branching defects (unpublished). This suggests that mTORC2 is not a major player in controlling lung branching. Even if loss of mTORC1 results in increased mTORC2, it is unlikely that mTORC2 activation underlies the lung defects in mTORC1 mutants. It was reported that active mTOR signaling in the lung does not lead to branching defects (e.g., Saito et al. *Am J Respir Cell Mol Biol*, 2020). Consistent with this, loss of *Tsc1* or *Tsc2* (negative regulators of mTOR) in the lung epithelium using *Sox9-Cre* did not lead to branching defects (unpublished). Thus, increased mTORC2 in *Rptor* mutants is not expected to contribute to branching defects. This point has been clarified in the revision.

Is there enough detail provided in the methods for the work to be reproduced?

Yes

Reviewer #3 (Remarks to the Author):

Zhang et al NC review March 2022 DR

Zhang et al. knocked out Raptor in the mouse lung epithelium using ShhCre. Found defects in branching morphogenesis, but no changes in the trachea or main stem bronchus. No new daughter branches formed, and the mice died after birth with respiratory failure. Sox2+ cells were absent from the 5 main branches that arise from the primary buds at ~E11.5 and remained confined to the trachea and main stem bronchi as the Raptor KO mice age. Sox2+ cells normally differentiate from Sox9+ cells, with Sox9 giving rise to AT1/2 cells and Sox2 giving rise to goblet/club/ciliated cells in the proximal airway. Therefore, these results suggest that Sox9 progenitors require mTORC1 to produce Sox2+ cells, resulting in loss of conducting airways in Raptor-KO lungs. To understand the mechanisms through which mTORC1 controls lung branching, the authors studied mitochondria and found a decrease in whole lung mtDNA/nDNA ratio of about 30% (Fig 5D,E) and ~25% less ATP production. To test whether this is the mechanism in vivo, they knocked out Tfam and Cox10 using ShhCre. The MtDNA/nDNA ratio was decreased by ~50% in the Tfam KO mice, as expected, and ATP production was decreased by ~40% in both models. Defects in branching morphogenesis were found in these mice, similar to the Raptor KO, and similar defects in the distribution of Sox2+ cells. Saccule formation by the Sox9+ cells appeared to be intact.

It is clear from the data that mTORC1 plays a crucial role in branch formation within the developing lung, and that that mitochondrial DNA and ATP production are decreased in the raptor knockout lung, but there is not strong evidence to link these two observations. While the mice with impaired mitochondria develop a similar lung phenotype to the raptor KO mice, this similarity does not imply causation. In addition to this concern, many of the data rely on IF images. Quantitative metrics of the imaging data are needed along with protein and mRNA analyses to validate the microscopic observations.

Major comments:

1) The term “tubular organ” in the title is an unusual choice – replace with developing murine lung? Also in the title, it cannot be concluded based on the data that mTORC1 is acting via mitochondrial capacity – this an interesting hypothesis but not proven by the available data.

As suggested by the reviewer, we have replaced “tubular organ” with “developing murine lung” in the title.

We have purified lung epithelial cells using anti-EpCAM antibody from control and *Rptor* mutant lungs and conducted quantitative analyses. They include a reduction in the ratio of mitochondrial DNA to nuclear DNA (new Fig. 5d, e) and the relative ATP production (new Fig. 5f). This is consistent with a model in which mTORC1 controls mitochondrial capacity and ATP production. Moreover, reduced ATP production in *Tfam*-, *Cox10*- and *Lrpprc*-deficient lungs also led to lung defects similar to those in *Rptor* lungs. Together, these findings support the notion that mTORC1 controls mitochondrial capacity.

mTORC1 regulates multiple cellular events including mitochondrial function, metabolism, protein synthesis and autophagy. Several targets (e.g., 4E-BP, S6K and ULK) of mTORC1 have been identified. It is interesting to note that disruption of *4e-bp1*, *4e-bp2*, *S6K1* and *Ulk1/2* in mice does not lead to branching defects in the lungs as reported in the literature (e.g., Le Bacquer et al. *J Clin Invest* 2007; Shima et al. *EMBO J*, 1998; Cheong et al. *Autophagy*, 2014). Thus, genetic studies are required to identify the functional targets of mTORC1 *in vivo*. Our results in this study suggest that mTORC1 controls mitochondrial capacity during lung branching.

2) While the IF data are beautiful and informative, the lack of western blot and qPCR analyses is surprising. The authors could take advantage of the mTmG system to sort their tissues by FACS. They could have used FACS to isolate GFP+, CRE+ cells prior to western blotting, qPCR, or even RNAseq.

As mentioned above, we have purified lung epithelial cells using anti-EpCAM antibody from control and *Rptor*, *Tfam* and *Cox10* mutant lungs and conducted quantitative analyses. They include the relative ratio of mitochondrial DNA to nuclear DNA (new Fig. 5d, e and Fig. 6e, f), relative ATP production (new Fig. 5f and Fig. 6g) and Western blotting analysis of pathway components (new Supplementary Fig. 1c and Supplementary Fig. 10a).

3) A more quantitative approach to analyze the lung development phenotypes would strengthen the conclusions considerably. Currently, the data are largely qualitative. How many individual mice were dissected and immuno-stained? Did all the mice show similar development at each timepoint and what was the range of variation between mice of the same genotype? What measures were taken to ensure that images were taken from the same exact region in control and *raptor fl/fl* lungs? Were any measurements taken of lung mass or volume? Perhaps the authors could quantitate the length and number of lung branches at each time point.

We have conducted more rigorous quantifications on control and mutant lungs. They are shown in new Fig. 1b, d, g, h, Fig. 2b, Fig. 4e-i, Fig. 5h, j, Fig. 6c-d, Fig. 8c-e, Supplementary Fig. 1a, b, Supplementary Fig. 3b, d, Supplementary Fig. 6e, Supplementary Fig. 7b, d, f, Supplementary Fig. 8b, d, and Supplementary Fig. 9c.

We have included the number (n) of mice analyzed in the figure and figure legends. Of note, we have analyzed a far larger number of control and mutant lungs over a period of several years. The n number shown in the manuscript simply represents a fraction of the animals on which a

thorough analysis was conducted. Since an average of 2 mutants were obtained in each litter, we combined mutants from different litters for statistical analysis.

The mutant phenotypes described in this study were completely penetrant. All the mice of the same genotype showed similar lung phenotypes at each timepoint and there was little variation between the mutant lungs. The mutant phenotypes were apparent and could be discerned by the naked eye or under the dissecting scope. Accordingly, changes in morphology and molecular markers in the mutants can also be readily distinguished from the controls under the microscope. When we assessed the phenotypes using quantifiable parameters (*e.g.*, distance, diameter, thickness, area, relative signal density, cell type percentage), the range of variation between mice of the same genotype was less than 20%. This is apparent from the data points in all bar graphs.

The completely penetrant and invariant phenotypes have allowed us to provide representative images shown in the figures. It has also allowed us to obtain statistical significance with the number of mice used in this study.

Different regions of the lungs were characterized by molecular markers (*e.g.*, SOX2 and SOX9). This has enabled us to compare corresponding regions in control and mutant lungs despite structural alteration of the mutant lungs.

As suggested by the reviewer, we have measured lung mass by DNA volume and wet weight at 12.5 and 16.5 *days post coitus (dpc)* (new Supplementary Fig. 1a, b).

We did not quantitate the length and number of branches since lung branching ceased after 11.5 *dpc* in *Rptor* mutant lungs.

4) 1A-F: Is pRPS6 decreased in the raptor fl/fl mice, or are total levels decreased? Additional IF staining for total RPS6, and/or a western blot, are needed. The same is true for pMLC2 in figure S1.

We have provided quantification of the expression levels of total RPS6 and MLC2. The immunofluorescent signal of RPS6 is shown in new Fig. 1a, while MLC2 levels by Western blotting is shown in new Supplementary Fig. 1c.

5) 1X-K: Because the structure of these tissues varies so much between control and raptor fl/fl mice, IF alone cannot be used to evaluate expression of F-Actin and smooth muscle actin. If western blotting or qPCR are not possible, the authors might be able to normalize the fluorescent signal to total area of the tissue. This can be done in ImageJ.

As suggested by the reviewer, we have quantified phalloidin signal (F-actin) using ImageJ and normalized the signal to individual epithelial cells (new Fig. 1g). The levels of monomeric actin appeared to be similar between control and *Rptor* mutant lungs as judged by RNA-Seq and Western blotting analysis.

RNA-Seq analysis also revealed a reduction in *Acta2* (smooth muscle actin, SMA) levels in *Rptor* mutant lungs. We noted that the distance between the epithelial tip and mesenchymal SMA signal was increased in *Rptor* mutant lungs (new Fig. 1h). This suggests that the proximal-distal transition of the mesenchyme was also affected in the absence of epithelial *Rptor*.

6) 1S-W: How was the Edu stain quantified? Was the Edu signal normalized to DAPI signal?

The rate of epithelial proliferation was calculated as the ratio of EdU⁺ E-Cad⁺ DAPI⁺ positive cells to E-Cad⁺ DAPI⁺ cells. E-Cad marked all epithelial cells. DAPI helped identify individual cells. The EdU signal is nuclear and co-localizes with the DAPI signal. For each animal, at least five independent sections were used for counting. We have added more details of EdU quantification in the revised figure legends and the Methods section.

7) 2O-V: *The introduction states that cessation of Sox9 expression is concomitant with Sox2 activation, but there is significant overlap of SOX2 and SOX9 expression in these images.*

“Cessation of SOX9 expression is concomitant with SOX2 activation” is meant to be a general statement. In detail, there is a transition region where SOX9 expression levels are significantly reduced and SOX2 expression levels are on the rise. Also, the background levels of SOX9 immunostaining may give the impression that SOX9 expression overlaps with SOX2 expression. To better visualize SOX2 and SOX9 expression, we have provided new images with a different exposure (new Fig. 2d) and a different set of immunostaining that allows for a better characterization of SOX9/SOX2 segregation (new Supplementary Fig. 4a).

8) 2W-H': *Quantification is necessary to conclude that the Sox9 -> Sox2 transition is impaired in raptor fl/fl mice. Perhaps a ratio of RFP signal to GFP signal, normalized to total area of the lungs, could be used.*

As suggested by the reviewer, we have measured the area of SOX2⁺ and SOX9⁺ signals, normalized to the total area at 11.5, 12.5 and 13.5 dpc (new Fig. 2b).

9) 3A-H: *please include the RFP channel alone*

Images of the RFP channel have been provided (new Fig. 3a, b and Supplementary Fig. 5a).

10) 4: *Please provide quantification of the fluorescent signal relative to DAPI.*

We have provided quantification of the fluorescent signal in Figure 4. The fluorescent pRPS6 signal was measured using ImageJ, while the number of epithelial cells was counted by DAPI (new Fig. 4i). The number of SOX2⁺, Ac-tub⁺, CC10⁺, HOPX⁺ and SPC⁺ cells were counted via marker expression for each cell type; the total number of epithelial cells was determined by DAPI (new Fig. 4e, f, g, h).

11) 5A,B,C: *The authors chose to focus on the difference in mitochondrial content activity between control and raptor fl/fl mice, which is surprising given that no mitochondrial pathways came up in the GO analyses. This should be discussed.*

We previously reported regulation of mitochondrial function by mTORC1 during alveolar formation (Zhang et al. *eLife*, 2022). We thus speculate that mTORC1 signaling also controls mitochondrial function during lung branching. This idea was confirmed by immunostaining of MPC1 and MTCO1, qPCR analysis of the mtDNA/nDNA ratio and measurement of ATP levels in control and *Rptor* mutant lungs.

In the revision, we have also performed RNA-Seq analysis of control, *Rptor^{fl/fl}*; *Shh^{Cre/+}*, *Tfam^{fl/fl}*; *Shh^{Cre/+}* and *Cox10^{fl/fl}*; *Shh^{Cre/+}* lungs at 14.5 dpc to try to identify common downstream targets

(new Fig. 9). The mitochondrial pathway was not among the top 40 enriched pathways (new Supplementary Fig. 15). This is likely because *Rptor* controls mitochondrial genes posttranscriptionally. This point has been clarified in the revision.

12) 5, 6, 8: It is unclear whether there is less MTOC expressed in each of the knockout models, relative to control mice. There is a decrease in GFP signal, but this decrease could be due to the increase in cystic “empty” space. For a proper comparison, quantification to GFP signal relative to tissue area, in multiple sections, should be performed.

We have quantified the MTCO1 signal in different regions of the lungs in *Rptor*, *Tfam* and *Cox10* mutant mice (new Figs. 5h, 5j and Figs. 6c, 6d). In all cases, the MTCO1 signal was reduced in the mutant lungs, consistent with the published work on other tissues and cell lines.

Minor comments:

1) Legend for Fig 6 A-P includes the presumed mechanisms through which MTCO1 is reduced in TFAM and Cox10 KO (reduced transcription of MTCO1; reduced MTCO1 protein stability, etc). Since the authors did not generate these data, this information should perhaps be in the manuscript itself and not the figure legend.

As suggested by the reviewer, we have moved the aforementioned description from the figure legend to the manuscript proper.

2) Legend for Fig 9A – progenies should be progeny.

We have corrected this mistake.

3) Model, Figure 9A – is there evidence that mTORC1/mitochondria contribute to the differentiation of Sox9+ cells into AT1/AT2? There is a “thin” arrow here.

We do not have definitive evidence to show that mTORC1/mitochondria control the differentiation of SOX9⁺ cells to AT1/AT2. The thin arrow has been removed in the revised model.

4) Legend for 9B – what is meant by “in contrast” – in contrast to what?

We have removed “in contrast” in the revision.

5) For the quantification of EdU positive cells (Fig 1W and elsewhere) how was the number of EdU positive cells measured? This does not seem to be in the legend or methods.

The number of EdU⁺ cells were counted manually using ImageJ. The detailed procedure of EdU quantification has been included in the revised Methods section.

6) Figure S6: typo in title (reduction misspelled)

The typo in Figure S6 (now Supplementary Fig. 9) has been corrected.

REVIEWERS' COMMENTS

Reviewer #1 (Remarks to the Author):

The authors have performed further experiments aiming to address this reviewer's concerns. They provided RNA sequencing and quantification data that further support the important role of mTOR signaling in lung development.

Reviewer #2 (Remarks to the Author):

The authors have address the main concerns from the first round of reviews and the revised manuscript uses innovative methods to provide new insight into the role of mTORC signalling in airway branching morphogenesis. The study makes a valuable contribution in developmental biology.

Reviewer #3 (Remarks to the Author):

The authors have done a beautiful job addressing the many comments. This has strengthened the manuscript considerably. The only remaining concern relates to comment #1 and the title of the manuscript. The data presented support an association with mitochondrial capacity, but cannot be used to support the conclusion that mTORC1 is acting via mitochondrial capacity (a causal relationship), as indicated by the title. The authors clearly acknowledge this by using the term "suggest" in the rebuttal "our results in this study suggest that mTORC1 controls mitochondrial capacity during lung branching" and the abstract "This suggests that mTORC1 promotes differentiation of SOX9+ progenitors to form the conducting airways by modulating mitochondrial capacity." The title could be changed to "mTORC1-dependent facilitation of differential stem cell differentiation to shape the developing murine lung is associated with mitochondrial capacity" to better reflect the data.

Reviewer #1 (Remarks to the Author):

The authors have performed further experiments aiming to address this reviewer's concerns. They provided RNA sequencing and quantification data that further support the important role of mTOR signaling in lung development.

We greatly appreciate the reviewer's comments and suggestions for improving the manuscript.

Reviewer #2 (Remarks to the Author):

The authors have address the main concerns from the first round of reviews and the revised manuscript uses innovative methods to provide new insight into the role of mTORC signalling in airway branching morphogenesis. The study makes a valuable contribution in developmental biology.

We greatly appreciate the reviewer's comments and suggestions for improving the manuscript.

Reviewer #3 (Remarks to the Author):

The authors have done a beautiful job addressing the many comments. This has strengthened the manuscript considerably. The only remaining concern relates to comment #1 and the title of the manuscript. The data presented support an association with mitochondrial capacity, but cannot be used to support the conclusion that mTORC1 is acting via mitochondrial capacity (a causal relationship), as indicated by the title. The authors clearly acknowledge this by using the term "suggest" in the rebuttal "our results in this study suggest that mTORC1 controls mitochondrial capacity during lung branching" and the abstract "This suggests that mTORC1 promotes differentiation of SOX9+ progenitors to form the conducting airways by modulating mitochondrial capacity." The title could be changed to "mTORC1-dependent facilitation of differential stem cell differentiation to shape the developing murine lung is associated with mitochondrial capacity" to better reflect the data.

We also thank the reviewer's tremendous effort in helping us improve the manuscript.

As suggested by the reviewer, we have replaced "by modulating" with "and is associated with" in the tile. The new title is: "mTORC1 signaling facilitates differential stem cell differentiation to shape the developing murine lung and is associated with mitochondrial capacity".